# Norm×Direction: Restoring the Missing Query Norm in Vision Linear Attention

**Weikang Meng** [1 2]  **Yadan Luo** [3]  **Liangyu Huo** [1]  **Yingjian Li** [2]  **Yaowei Wang** [1 2]  **Xin Li** [2]  **Zheng Zhang** [1]

## Abstract

Linear attention mitigates the quadratic complexity of softmax attention but suffers from a critical loss of expressiveness. We identify two primary causes: (1) The normalization operation cancels the query norm, which breaks the correlation between a query's norm and the spikiness (entropy) of the attention distribution as in softmax attention. (2) Standard techniques for enforcing non-negativity cause destructive information loss by nullifying valid inner-product interactions. To address these challenges, we introduce **NaLaFormer**, a novel linear attention mechanism built upon a norm×direction (ND) decomposition of the query and key vectors. We leverage each component to solve a distinct problem: The *query norm* is injected into our kernel to create a query-norm-aware map that restores the attention distribution's spikiness. The *direction vectors* are processed by a geometric, cosine-based similarity metric that guarantees non-negativity while preserving the rich, fine-grained information of the inner product. We validate NaLaFormer through a comprehensive multi-modal evaluation, where it sets new state-of-the-art benchmarks for linear attention. Our model achieves up to a 7.5% accuracy gain on ImageNet-1K and a 4.7% mIoU improvement on ADE20K over comparable baselines. It demonstrates profound efficiency, reducing peak memory by a transformative 92.3% in token-intensive super-resolution tasks (70K+ tokens). NaLaFormer's versatility is further confirmed as it surpasses strong baselines like Mamba on common-sense reasoning and sets a new state-of-the-art on the Long Range Arena (LRA) benchmark. Code is available at this https URL.

## 1. Introduction

Transformer models (Vaswani et al., 2017; Dosovitskiy et al., 2021) have demonstrated remarkable success in both vision and language tasks. The core self-attention mechanism models global contextual relationships through softmax-normalized dot-product similarity, but incurs quadratic complexity $\mathcal{O}(N^2)$ relative to sequence length $N$, creating significant computational overhead for long sequences or high-resolution images. To address this limitation, linear attention (Katharopoulos et al., 2020; Cai et al., 2023; Han et al., 2023; MiniMax et al., 2025; Lu et al., 2024a) replaces the $\exp(\cdot)$ operator in softmax with a linearly separable kernel $\phi(\cdot)$. This reformulation reorders computation priorities from $\exp(\mathbf{q}_i \mathbf{k}_j^\top)\mathbf{v}_j$ to $\phi(\mathbf{q}_i)(\phi(\mathbf{k}_j)^\top \mathbf{v}_j)$ achieving linear complexity through associative matrix multiplication.

Although linear attention mechanisms have gained popularity for their efficiency in sequence modeling, yet they consistently *underperform* compared to their softmax-based counterparts. A central limitation lies in the restricted expressiveness of the kernel function $\phi(\cdot)$, which approximates attention through inner products of transformed queries and keys, $\phi(\mathbf{q})^\top \phi(\mathbf{k}_i)$. Early approaches focused on ensuring *non-negativity*, a necessary condition for interpreting attention scores as normalized distributions. To this end, various activation functions have been employed, including ReLU (Han et al., 2023; Cai et al., 2023), $1 + \text{ELU}$ (Katharopoulos et al., 2020), SiLU (Yang et al., 2024b; MiniMax et al., 2025), reorientation activation (Meng et al., 2026), as well as positive-valued randomized feature mappings such as the Gaussian kernel (Lu et al., 2024a; Chen et al., 2021). However, these kernels inherently discard negative components of the input, limiting their ability to capture the full range of semantic relationships.

As a result, linear attention often yields overly smooth attention distributions, lacking the *spikiness* characteristic of softmax attention. This leads to elevated entropy and hinders the model's ability to focus on semantically critical tokens. Recent efforts such as Hedgehog (Zhang et al., 2024), FLatten Transformer (Han et al., 2023), and PolaFormer (Meng et al., 2025), attempt to address this shortcoming by introducing element-wise power functions to sharpen token-wise attention. While these methods empirically improve discrimination, the underlying cause of the entropy explosion in linear attention remains poorly understood.

---

[1]Harbin Institute of Technology, Shenzhen, China [2]Pengcheng Laboratory, China [3]The University of Queensland, Australia. Correspondence to: Zheng Zhang <darrenzz219@gmail.com>.

*Proceedings of the 43rd International Conference on Machine Learning*, Seoul, South Korea. PMLR 306, 2026. Copyright 2026 by the author(s).

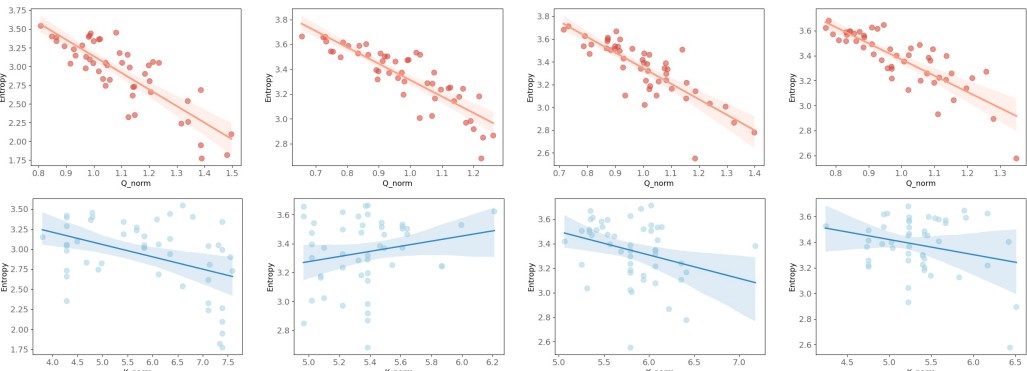

*Figure 1.* **Entropy-norm correlation in softmax attention.** We plot the relationship between feature entropy and vector norms in a Swin Transformer sampled on ImageNet. The top row shows q-norms ($x$-axis) exhibit a strong negative correlation with attention entropy ($y$-axis). The bottom row shows that k-norms have no consistent effect. This observation suggests that the entropy diminishing in linear attention may stem from insufficient query scaling, pointing to the key for restoring spikiness.

To further explore the property of entropy, inspired by prior studies (Dehghani et al., 2023; Naseer et al., 2021; Milakov & Gimelshein, 2018) that have identified the $\mathbf{q}\mathbf{k}^\top$ norm cancellation in softmax attention, we notice that linear attention exhibits a different behavior, especially showing *asymmetric sensitivity* to the query and key norms. To analyze this effect, we employ a norm×direction (ND) decomposition of linear attention:

$$\text{LinearAttn}_t = \frac{\phi(\mathbf{q}_t) \sum_{i=1}^{N} \phi(\mathbf{k}_i)^\top \mathbf{v}_i}{\phi(\mathbf{q}_t) \sum_{j=1}^{N} \phi(\mathbf{k}_j)^\top} =$$

$$\frac{\underbrace{||\phi(\mathbf{q}_t)||}\ \text{dir}(\phi(\mathbf{q}_t))\ \sum_{i=1}^{N} ||\phi(\mathbf{k}_i)||\ \text{dir}(\phi(\mathbf{k}_i))^\top\ \mathbf{v}_i}{\underbrace{||\phi(\mathbf{q}_t)||}_{\text{q}-\text{norm}-\text{unaware}}\ \text{dir}(\phi(\mathbf{q}_t)) \sum_{j=1}^{N} ||\phi(\mathbf{k}_j)||\ \text{dir}(\phi(\mathbf{k}_j))^\top}.$$

$$(1)$$

where $\text{dir}(\mathbf{x}) = \mathbf{x}\ /\ ||\mathbf{x}||$ refers to the direction component. Eq. (1) exposes a critical *asymmetry* where only *key norms* influence linear attention outputs, as *query norms* are reduced through normalization. To further test this conjecture, Fig. 1 demonstrates a strong inverse correlation between entropy and query norms in softmax attention, whereas key norms exhibit weak correlation with spikiness. Notably, current linear attention approaches utilize element-wise kernel functions to enforce non-negativity constraints, suffering from the norm degradation and negative values loss.

In this work, we establish a mathematical framework characterizing query norm-entropy control in softmax attention. Based on these insights, we propose **Norm-aware Linear Attention**, a novel mechanism that explicitly couples $||\phi(\mathbf{q}_t)||$ with spikiness to address the limitation of the query norm unawareness in linear attention. Our theoretical analysis reveals the dynamic control of entropy reduction (spikiness) from $||\phi(\mathbf{q}_t)||$. Specifically, for each direction of $\mathbf{q}_t$, the entropy decreases with a great $||\mathbf{q}_t||$ monotonically. Empirical validation through randomized sampling of attention computations (Fig. 1 and Fig. 2 (b)) demonstrates that the $||\mathbf{q}_t||$ is practically great enough in most cases. To jointly

preserve spikiness and norm awareness, we employ a power function for each $\mathbf{q}_t$ with an adaptive query norm aware power. To address norm degradation in conventional linear attention while preserving non-negativity constraints, we proposed a *cosine direction similarity* algorithm to only map the direction component. Utilizing Ptolemy's theorem as a geometric foundation, our method employs cosine similarity for dimensional rescaling, selectively suppressing dimensions with distant directions and keeping closed dimensions. These synergistic innovations faithfully capture essential properties of softmax operators while maintaining computational efficiency.

We establish NaLaFormer's state-of-the-art performance and broad applicability through a rigorous multi-modal evaluation spanning foundational vision benchmarks, including image classification, object detection, semantic segmentation, and highly challenging long-sequence scenarios. Our model sets new performance standards for linear attention, achieving up to a **7.5%** accuracy gain over comparable models on ImageNet-1K and a **4.7%** mIoU improvement on ADE20K. The architecture's advantages are particularly evident in token-intensive tasks: in super-resolution, which generates extremely long token sequences (over 70K tokens) from high-resolution images, NaLaFormer achieves a **92.3%** reduction in peak memory while cutting latency by **36.4%**. This long-sequence capability is further validated on the Long Range Arena (LRA) benchmark with **61.2%** average accuracy. Finally, to verify its versatility, we train a 340M-parameter language model from scratch, which surpasses strong autoregressive baselines like Mamba, establishing NaLaFormer as a powerful and efficient foundation for diverse modalities.

## 2. Preliminaries

An attention mechanism's efficacy is rooted in its ability to model the relationship between vectors. We posit that this relationship is defined by a fundamental duality of in-

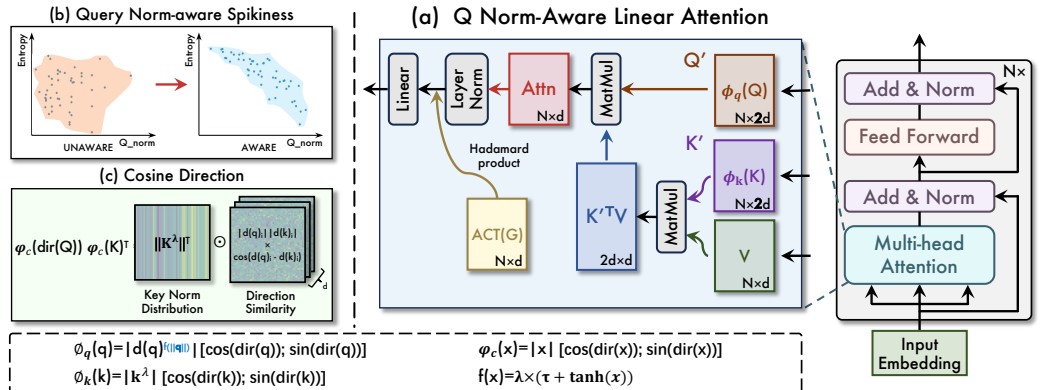

*Figure 2.* **The NaLaFormer architecture and its core mechanisms. (a)** The NaLaFormer block incorporates a simplified GLA and custom feature maps $\phi_q$ and $\phi_k$. **(b)** Our norm-aware method (right) restores the negative query norm-entropy correlation lost in standard linear attention (left). **(c)** The cosine direction mechanism enforces non-negativity by decomposing similarity into norm and direction components, preventing information loss.

formation: a vector's **norm**, which signals its importance, and its **direction**, which encodes its semantic orientation. An ideal attention mechanism must jointly leverage both. In this section, we employ this view to explain a failure in linear attention that underlies its gap to softmax attention.

To formalize our analysis, we first mathematically define the Norm×Direction (ND) decomposition:

---

**Definition 2.1** (ND Decomposition). Let $\mathbf{x} = (x_1, \ldots, x_d) \in \mathbb{R}^d$ is a non-zero vector, then the ND decomposition of $\mathbf{x}$ with $p$-norm is defined by:

$$\mathrm{ND}(\mathbf{x}; p) = ||\mathbf{x}||_p \cdot \mathrm{dir}(\mathbf{x}),$$

$$\text{where } \mathrm{dir}(\mathbf{x}) = \frac{(x_1, \ldots, x_d)}{||\mathbf{x}||_p}.$$

---

We name $\mathrm{dir}(\mathbf{x})$ the direction components of vector $\mathbf{x}$. According to the Norm Equivalence Theorem (Brezis, 2011), all norms on a finite-dimensional vector space are equivalent, thus we do not distinguish between different $p$-norm in the following discussions for simplicity.

### 2.1. Softmax Attention with ND Decomposition

Let $\mathbf{X} \in \mathbb{R}^{N \times D}$ denote a sequence of $N$ tokens with dimension $D$. We divide the dimension into $h$ heads, and each single head has $d$ dimensions. In a single head, the output $\mathbf{O} = \{\mathbf{o}_t\}_{t=1}^N \in \mathbb{R}^{N \times d}$ is computed following:

$$\mathbf{O} = \mathrm{Softmax}(\frac{\mathbf{Q}\mathbf{K}^\top}{\sqrt{d}})\mathbf{V}, \ \mathbf{o}_t = \frac{\sum_{i=1}^N \exp(\mathbf{q}_t \mathbf{k}_i^\top / \sqrt{d})}{\sum_{j=1}^N \exp(\mathbf{q}_t \mathbf{k}_j^\top / \sqrt{d})}\mathbf{v}_i,$$
(2)

in which $\mathbf{Q}, \mathbf{K}, \mathbf{V} \in \mathbb{R}^{N \times d}$ denote query, key and value vectors respectively with $N$ sequence length. The complexity of softmax attention is $\mathcal{O}(N^2 d)$. Then, we rewrite Eq. (2) with the ND-decomposition, which shows an explicit relation to query and key norms.

$$\mathbf{o}_t = \sum_{i=1}^N \frac{\exp(\mathbf{q}_t \mathbf{k}_i^\top / \sqrt{d}) \ \mathbf{v}_i}{\sum_{j=1}^N \exp(\mathbf{q}_t \mathbf{k}_j^\top / \sqrt{d})} =$$
$$\sum_{i=1}^N \frac{\exp(||\mathbf{q}_t|| \ ||\mathbf{k}_i|| \ \langle \mathrm{dir}(\mathbf{q}_t), \mathrm{dir}(\mathbf{k}_i) \rangle / \sqrt{d}) \ \mathbf{v}_i}{\sum_{j=1}^N \exp(||\mathbf{q}_t|| \ ||\mathbf{k}_j|| \ \langle \mathrm{dir}(\mathbf{q}_t), \mathrm{dir}(\mathbf{k}_j) \rangle / \sqrt{d})}.$$
(3)

This derivation reveals a critical property of softmax attention: the $\mathbf{Q}$-norm $||\mathbf{q}_t||$ is **preserved** within the exponential function, in stark contrast to its *cancellation* in linear attention as shown in Eq. (1). This allows the query norm to naturally act as a temperature score, where a larger query norm sharpens the attention distribution and reduces its entropy. The collapse of $\mathbf{Q}$-norm in conventional linear attention represents a fundamental departure from the softmax mechanism and potentially is a primary source of its diminishing entropy and performance drop.

## 3. Method

To mitigate this gap, we introduce NaLaFormer, which asymmetrically reformulates the linear attention kernel through the lens of ND decomposition. To achieve this, we explicitly restore the previously neglected **query norm** by integrating it into the query kernel map to dynamically regulate attention entropy (Sec. 3.1). Concurrently, we geometrically transform the decomposed **direction vectors** via a cosine similarity metric to guarantee non-negativity without the severe information loss of prior methods (Sec. 3.2). A rigorous theoretical proof can be found in Appendix A.1.

### 3.1. Query-Norm-Aware Feature Map

Linear attention (Katharopoulos et al., 2020) reformulates the $\mathbf{q}, \mathbf{k}$ similarity measure through linearly separable kernel function mappings $\mathrm{SM}(\mathbf{q}, \mathbf{k}) = \phi(\mathbf{q})\phi(\mathbf{k})^\top$, where the feature map $\phi(\cdot) : \mathbb{R}^d \to \mathbb{R}^{d'}$ is applied to query and key vectors. This allows for a reordering of computations, leading to an output formulation:

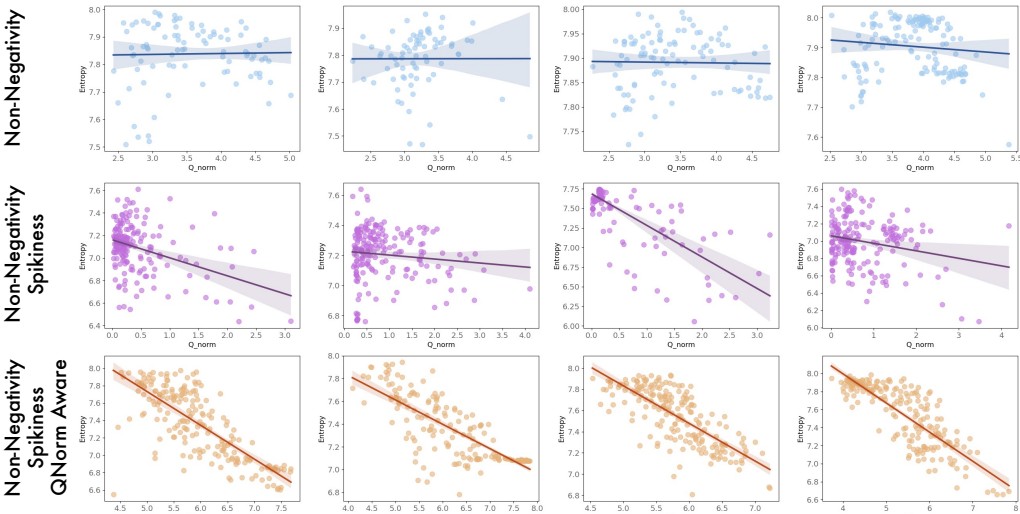

*Figure 3.* We visualize the query norm-entropy relationship under three approaches: (1) Only preserve non-negativity with $1 + \text{ELU}$ operator (Katharopoulos et al., 2020). (2) Keep both non-negativity and spikiness with ReLU operator and power function as in FLatten (Han et al., 2023). (3) Our q-norm-aware approach shows a clear correlation between entropy and query norms.

$$\mathbf{o}_t = \sum_{i=1}^{N} \frac{\phi(\mathbf{q}_t)\phi(\mathbf{k}_i)^\top \mathbf{v}_i}{\sum_{j=1}^{N} \phi(\mathbf{q}_t)\phi(\mathbf{k}_j)^\top} = \frac{\phi(\mathbf{q}_t)\sum_{i=1}^{N}\phi(\mathbf{k}_i)^\top \mathbf{v}_i}{\phi(\mathbf{q}_t)\sum_{j=1}^{N}\phi(\mathbf{k}_j)^\top} =$$

$$\frac{\|\phi(\mathbf{q}_t)\| \, \text{dir}(\phi(\mathbf{q}_t)) \, \sum_{i=1}^{N} \|\phi(\mathbf{k}_i)\| \, \text{dir}(\phi(\mathbf{k}_i))^\top \, \mathbf{v}_i}{\underbrace{\|\phi(\mathbf{q}_t)\|}_{\text{q-norm-unaware}} \, \text{dir}(\phi(\mathbf{q}_t)) \sum_{j=1}^{N} \|\phi(\mathbf{k}_j)\| \, \text{dir}(\phi(\mathbf{k}_j))^\top}.$$

$$(4)$$

Here, the cancellation of $\|\phi(\mathbf{q}_t)\|$ reveals that the mainstream linear attention is "**query-norm unaware**". As shown in Fig. 3 (row 2), this causes a *breakdown* of the entropy-norm correlation observed in softmax attention. Despite existing approaches (Meng et al., 2025; Han et al., 2023; 2024c) leverages exponential functions to reduce entropy, they remain insensitive to query norm.

Motivated by this, we therefore design the **query-norm-aware** feature map to explicitly encode the query's norm into its feature map:

$$\varphi_q(\mathbf{q}) = \text{dir}(\mathbf{q})^{\text{f}(\|\mathbf{q}\|)}, \; \varphi_k(\mathbf{k}) = \mathbf{k}^\lambda, \quad (5)$$

where $\text{f}(x) = \lambda * (\tau + \tanh(x))$ serves as a *norm-dependent sharpening* function, which dynamically modulates the sequence entropy of the attention to restore the sharpness characteristics of standard softmax attention. A full theoretical analysis of this property is provided in Sec. A.1. For simplicity and to ensure a fair comparison with other power-based counterparts (Han et al., 2023), we apply the power function for rescaling (Fig. 3). The hyperparameters $\lambda, \tau$ are used to constrain the exponent's range, ensuring it remains greater than one while also preventing numerical overflow.

**Empirical Observations.** With a one-line modification, we restore the negative correlation between query norm and entropy that is lost in linear attention. To validate this,

we again visualize the entropy–norm relationship in linear attention under three feature maps using the same inputs and layers (Fig. 3). The first row shows the simplest linear attention with $\text{ReLU}(\cdot)$, which yields relatively high entropy and no clear correlation. The second row depicts the *power-based* Flatten (Han et al., 2023) Transformer's $\text{ReLU} + \text{power}$ mapping, which reduces entropy and sharpens token distinctions, yet the entropy–norm correlation remains inconsistent with softmax (Fig. 1). In the third row, we incorporate the query norm into the power function factor. Notably, this modification restores the negative correlation between query norm and entropy, demonstrating that our method successfully preserves the negative correlation between query norm and entropy in softmax attention.

### 3.2. Keep Non-Negativity with Cosine Direction

A second challenge in linear attention design is enforcing *non-negativity*. Prevalent methods achieve this by adopting $\text{ReLU}(\cdot)$ and $1 + \text{ELU}(\cdot)$ to suppress negative values element-wise. However, these approaches incur *destructive* information loss, as it nullifies any interactions where $\mathbf{q}_i \mathbf{k}_i < 0$. This often leads to a sparse and less informative similarity representation. To address this, we propose a structure-preserving alternative based on a trigonometric isomorphism and define a mapping $\varphi_c(\cdot)$ that transforms each scalar direction component $\text{dir}(\mathbf{q})_i$ and $\text{dir}(\mathbf{k})_i$ into a 2D vector:

$$\varphi_c(\text{dir}(\mathbf{q})_i) = \begin{pmatrix} |\text{dir}(\mathbf{q})_i| \cos(\text{dir}(\mathbf{q})_i) \\ |\text{dir}(\mathbf{q})_i| \sin(\text{dir}(\mathbf{q})_i) \end{pmatrix},$$

$$\varphi_c(\text{dir}(\mathbf{k})_i) = \begin{pmatrix} |\text{dir}(\mathbf{k})_i| \cos(\text{dir}(\mathbf{k})_i) \\ |\text{dir}(\mathbf{k})_i| \sin(\text{dir}(\mathbf{k})_i) \end{pmatrix}. \quad (6)$$

This formulation elegantly decouples magnitude from sign, encoding their interaction through the cosine of their angular

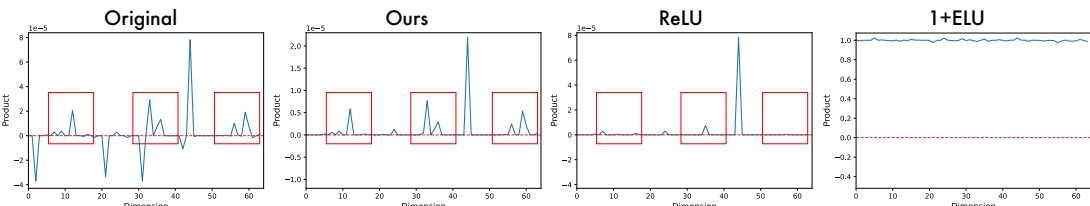

*Figure 4.* **Comparisons of element-wise dot product contributions for different non-negative strategies.** The plots show $\mathbf{q}_i\mathbf{k}_i$ value for (1) the raw inputs, (2) our novel cosine-based approach, (3) ReLU activation, and (4) $1 + $ ELU activation. Our approach ensures all dimensional contributions are non-negative while retaining the fine-grained "spikiness" observed in the original product.

difference. The sum of inner products of these corresponding 2D vectors for query $\mathbf{q}$ and key $\mathbf{k}$ for each dimension $i$ then becomes,

$$
\begin{aligned}
&\sum_{i=1}^{d} \varphi_c(\mathrm{dir}(\mathbf{q}))_i \varphi_c(\mathrm{dir}(\mathbf{k}))_i^{\top} \\
&= \sum_{i=1}^{d} |\mathrm{dir}(\mathbf{q})_i||\mathrm{dir}(\mathbf{k})_i|(\cos(\mathrm{dir}(\mathbf{q})_i)\cos(\mathrm{dir}(\mathbf{k})_i) \\
&\quad + \sin(\mathrm{dir}(\mathbf{q})_i)\sin(\mathrm{dir}(\mathbf{k})_i)), \\
&= \sum_{i=1}^{d} |\mathrm{dir}(\mathbf{q})_i||\mathrm{dir}(\mathbf{k})_i|\cos(\mathrm{dir}(\mathbf{q})_i - \mathrm{dir}(\mathbf{k})_i).
\end{aligned} \quad (7)
$$

Due to the properties of trigonometric functions, we have $\cos(x)^2 + \sin(x)^2 = 1$, thus, $\|\varphi_c(\cdot)\|$ is constant, which keeps the norm of the direction vector fixed. To ensure the cosine term being non-negative, we scale each elements of the direction components into $[-\frac{\pi}{4}, \frac{\pi}{4}]$ with a tanh-based mapping $\tanh(\frac{\mathbf{x}}{\|\mathbf{x}\|}) \times \frac{\pi}{4}, x \in \{\mathrm{dir}(\mathbf{q}), \mathrm{dir}(\mathbf{k})\}$. This guarantees that the resulting angle difference remains within the interval $(\mathrm{dir}(\mathbf{q})_i - \mathrm{dir}(\mathbf{k})_i) \in [-\frac{\pi}{2}, \frac{\pi}{2}]$.

**Empirical Observations.** The benefit of this information-preserving approach is empirically evident in Fig. 4, which shows the dimensional results of the dot products between the $\mathbf{q}$ and $\mathbf{k}$ vectors under different non-negativity-preserving feature maps. Compared with the original $\mathbf{q}_i\mathbf{k}_i$ dot product, prior approaches that rely on ReLU to enforce non-negativity discard a significant amount of spikiness information during the inner-product computation. Methods based on $1 + $ ELU not only lose spikiness but also exhibit reduced discriminability across dimensions. In contrast, our proposed method effectively overcomes these limitations and is able to retain richer information.

**Differences from Previous Works.** Existing works, such as Cosformer (Qin et al., 2022) and RoPE (Su et al., 2024), keep part of the information with trigonometric functions by using cosine-based functions. Cosformer replaces the softmax in attention with a cosine-based distance metric, using cosine similarity to directly measure query-key alignment, while RoPE encodes absolute positions via rotation matrices in complex space to represent the relative position. However, both kinds of cosine similarity are employed for **positional decay**, which differs from our cosine inhibition method targeting the **similarity and dimensions with opposite signals**, shown as follows:

$$
\mathrm{SM}_{\mathrm{cosformer}}(\mathbf{q}_n, \mathbf{k}_m) = \phi(\mathbf{q_n})\phi(\mathbf{k_m})^{\top} \underbrace{\cos(\frac{(m-n)\pi}{2M})}_{\text{relative position}},
$$
$$
\quad (8)
$$

$$
\mathrm{SM}_{\mathrm{rope}}(\mathbf{q}_n, \mathbf{k}_m) = \phi(\mathbf{q_n})\ \underbrace{R^d_{\Theta, n-m}}_{\text{relative position}}\ \varphi(\mathbf{k_m})^{\top}, \quad (9)
$$

$$
\mathrm{SM}_{\mathrm{ours}}(\mathbf{q}, \mathbf{k}) = \sum_{i=1}^{d} \underbrace{\cos(\phi(\mathbf{q})_i - \phi(\mathbf{k})_i)}_{\text{dimensional cosine similarity}} \quad (10)
$$

### 3.3. NaLaFormer: A Unified Norm-Aware Linear Attention

We now synthesize our principles of norm-awareness and non-destructive similarity into a unified model: **NaLaFormer**. This architecture is designed to concurrently restore the critical query-norm-entropy correlation of softmax attention while preserving dimensional information typically lost in other linear attention variants. This is achieved by designing feature maps $\phi_q(\cdot)$ and $\phi_k(\cdot)$ that integrate both norm- and direction-aware mappings *i.e.*, $\varphi_q$ and $\varphi_c$ aforementioned:

$$
\phi_q(\mathbf{q}) = [\phi_q^{\cos}(\mathbf{q}); \phi_q^{\sin}(\mathbf{q})],\ \phi_k(\mathbf{k}) = [\phi_k^{\cos}(\mathbf{k}); \phi_k^{\sin}(\mathbf{k})]. \quad (11)
$$

Each cosine and sine subcomponent's magnitude is carefully defined to be either norm-aware (for queries) or norm-scaled (for keys), while the direction is handled by our trigonometric mapping:

$$
\begin{aligned}
\phi_q^{\cos}(\mathbf{q}) &= |\mathrm{dir}(\mathbf{q})^{\mathrm{f}(\|\mathbf{q}\|)}|\cos(\mathrm{dir}(\mathbf{q})), \\
\phi_q^{\sin}(\mathbf{q}) &= |\mathrm{dir}(\mathbf{q})^{\mathrm{f}(\|\mathbf{q}\|)}|\sin(\mathrm{dir}(\mathbf{q})), \\
\phi_k^{\cos}(\mathbf{k}) &= |\mathbf{k}^{\lambda}|\cos(\mathrm{dir}(\mathbf{k})), \\
\phi_k^{\sin}(\mathbf{k}) &= |\mathbf{k}^{\lambda}|\sin(\mathrm{dir}(\mathbf{k})).
\end{aligned} \quad (12)
$$

Therefore, we can rewrite the outputs of linear attention as:

$$
\mathbf{S}_{\cos} = \sum_{i=1}^{N} \phi_k^{\cos}(\mathbf{k}_i)^{\top}\mathbf{v}_i, \mathbf{S}_{\sin} = \sum_{i=1}^{N} \phi_k^{\sin}(\mathbf{k}_i)^{\top}\mathbf{v}_i, \quad (13)
$$

and their corresponding normalization denominators as $\mathbf{Z}_{\cos} = \sum_{j=1}^{N} \phi_k^{\cos}(\mathbf{k}_j)^{\top}$ and $\mathbf{Z}_{\sin} = \sum_{j=1}^{N} \phi_k^{\sin}(\mathbf{k}_j)^{\top}$. Thus, the output $\mathbf{o}_t$ can be compactly formulated as:

$$
\mathbf{o}_t = \frac{\phi_q^{\cos}(\mathbf{q}_t)\mathbf{S}_{\cos} + \phi_q^{\sin}(\mathbf{q}_t)\mathbf{S}_{\sin}}{\phi_q^{\cos}(\mathbf{q}_t)\mathbf{Z}_{\cos} + \phi_q^{\sin}(\mathbf{q}_t)\mathbf{Z}_{\sin}} \odot \mathbf{G}. \quad (14)
$$

The NaLaFormer block integrates this attention mechanism within a gated architecture (Yang et al., 2024a; Qin et al., 2024). As shown in Fig. 2 (a), our norm-aware linear attention block first projects inputs to $\mathbf{Q}$, $\mathbf{K}$, $\mathbf{V}$ and then calculates $\text{LinearAttn}(\phi_q(\mathbf{Q}), \phi_k(\mathbf{K}), \mathbf{V})$, which then undergoes Layer Normalization. The output is subsequently modulated element-wise by a learned gate matrix $\mathbf{G}$ derived from the input, activated by SiLU, and finally passed through a linear layer to integrate the outputs from different heads. A rigorous theoretical analysis detailing how our norm-aware formulation systematically influences entropy reduction in both softmax and linear attention is provided in Appendix A.1 for completeness.

## 4. Experiments

In this section, we evaluate our **NaLaFormer** on various vision tasks. First of all, we conduct experiments on image classification on ImageNet-1K (Deng et al., 2009), object detection and instance segmentation on COCO (Lin et al., 2014), and semantic segmentation on ADE20K (Zhou et al., 2019) and CityScapes (Cordts et al., 2016), comparing performance with current efficient models. In addition, we conduct the Single Image Super-Resolution (SISR) task using DIV2K (Agustsson & Timofte, 2017) as the training dataset. For diffusion models, we integrate the proposed linear attention with DiT (Peebles & Xie, 2023) and SiT (Ma et al., 2024) using ImageNet-1K (Deng et al., 2009). To verify the generality of our method on language modality, we pre-train NaLaFormer language models from scratch and evaluate the pretrained model on common-sense reasoning tasks. At last, we assess NaLaFormer on the Long Range Arena (LRA) task (Tay et al., 2021) to compare against other linear attention models. Full experiment details and implementation details are provided in Appendix A.2, while ablation studies are reported in Appendix A.3.

### 4.1. Image Classification on ImageNet-1K

**Settings.** We train **NaLaFormer** from scratch on ImageNet-1K (Deng et al., 2009) using Top-1 accuracy. Consistent with established works (Fan et al., 2025b; 2024; Liu et al., 2021), we develop a set of NaLaFormer backbones, whose detailed configurations are summarized in Table 10. For fairness, we categorized baseline models into 5 classes according to their parameter sizes and FLOPs, then making performance comparisons within each group.

**Results.** As shown in Tab. 21, NaLaFormer consistently showing a higher accuracy comparing with the baseline models. For instance, NaLaFormer-T obtains an increase from 3.8% to 7.5% compared with baseline linear models with comparable FLOPS. Additionally, NaLaFormer-L consistently achieves a better performance compared with CNN, SSM and Transformer models. Notably, our model surpasses VRWKV-B (Duan et al., 2025) over 3.7% with fewer FLOPs. These results demonstrate NaLaFormer improves the expressive capability of the attention mechanisms through replacing the standard attention.

*Table 1.* Comparison with SOTA efficient vision models on ImageNet-1K. "Para", "FLOPs", and "Acc (%)" represent the number of parameters, FLOPs, and Top-1 accuracy, respectively.

| Model | Para | FLOPs | Acc |
|---|---|---|---|
| LocalVim-T (Huang et al., 2024) | 8M | 1.5G | 76.2 |
| MetaLA (Chou et al., 2024) | 6M | - | 75.3 |
| Mambaout-F (Yu & Wang, 2025) | 7M | 1.2G | 78.9 |
| EfficientVMamba-S (Pei et al., 2025) | 11M | 1.3G | 78.7 |
| NaLaFormer-XT | 8M | 1.0G | **79.1** |
| VAN-b1 (Guo et al., 2022c) | 14M | 2.5G | 81.1 |
| Conv2Former-N (Hou et al., 2024) | 15M | 2.2G | 81.5 |
| SBCFormer-L (Lu et al., 2024b) | 19M | 2.7G | 81.1 |
| RMT-T (Fan et al., 2024) | 14M | 2.5G | 82.4 |
| Agent-PVT-T (Han et al., 2024c) | 12M | 2.0G | 78.4 |
| NaLaFormer-T | 15M | 2.7G | **82.6** |
| MambaOut-T (Yu & Wang, 2025) | 27M | 4.5G | 82.7 |
| VMamba-T (Liu et al., 2024) | 30M | 4.9G | 82.6 |
| LocalVMamba-T (Huang et al., 2024) | 26M | 5.7G | 82.7 |
| Pola-Swin-T (Meng et al., 2025) | 29M | 4.5G | 82.6 |
| ViG-H-T (Liao et al., 2025) | 29M | 4.5G | 82.8 |
| MILA-T (Han et al., 2024b) | 25M | 4.2G | 83.5 |
| RAVLT-S (Fan et al., 2025b) | 26M | 4.6G | 84.2 |
| NaLaFormer-S | 26M | 5.1G | **84.3** |
| MambaOut-S (Yu & Wang, 2025) | 49M | 9.0G | 84.1 |
| VMamba-S (Liu et al., 2024) | 50M | 8.7G | 83.6 |
| StructViT-B-8-1 (Kim et al., 2024) | 52M | 12G | 84.3 |
| SOFT-L (Lu et al., 2024a) | 64M | 11G | 83.1 |
| Pola-Swin-S (Meng et al., 2025) | 50M | 8.7G | 83.6 |
| ViG-H-S (Liao et al., 2025) | 50M | 8.8G | 83.8 |
| NaLaFormer-B | 52M | 12G | **85.2** |
| MambaOut-S (Yu & Wang, 2025) | 85M | 16G | 84.2 |
| FLatten-Swin-B (Han et al., 2023) | 89M | 15G | 83.8 |
| Agent-Swin-B (Han et al., 2024c) | 88M | 15G | 84.0 |
| Pola-Swin-B (Meng et al., 2025) | 88M | 15G | 83.8 |
| VRWKV-B (Duan et al., 2025) | 94M | 18G | 82.0 |
| RMT-L (Fan et al., 2024) | 95M | 18G | 85.5 |
| InLine-Swin-B (Han et al., 2024a) | 88M | 15G | 82.0 |
| MILA-B (Han et al., 2024b) | 96M | 16G | 85.3 |
| ViG-H-B (Liao et al., 2025) | 89M | 16G | 84.2 |
| RAVLT-L (Fan et al., 2025b) | 95M | 16G | 85.5 |
| NaLaFormer-L | 95M | 18G | **85.7** |

### 4.2. Object Detection and Instance Segmentation

**Settings.** We conducted the object detection task using COCO (Lin et al., 2014). To systematically evaluate architectural compatibility, we independently integrated NaLaFormer as the backbone into Mask R-CNN (He et al., 2017) and RetinaNet (Lin et al., 2017). All experiments were conducted using ImageNet-1k pretrained weights following the evaluation strategy in FLatten (Han et al., 2023).

**Results.** We show the results in Tab. 2, our model surpasses the other baseline models across various frameworks. For example, our NaLaFormer-T tested on Mask R-CNN detectors with "1 ×" schedule achieves 47.6 $AP^b$ and 43.0 $AP^m$, outperforming some larger baselines. Results of the experiment with RetinaNet are shown in Appendix A.9.

### 4.3. Semantic Segmentation

**Settings.** In this section, we integrate our model into the semantic segmentation task on ADE20K (Zhou et al., 2019) and CityScapes (Cordts et al., 2016). Similarly, we adopt our

*Table 2.* Object detection and instance segmentation results on the COCO dataset using Mask R-CNN with $1\times$ and $3\times$ schedule.

| METHOD | PARA (M) | FLOPS (G) | MASK R-CNN $1\times$ | | | | | | MASK R-CNN $3\times$ | | | | | |
|---|---|---|---|---|---|---|---|---|---|---|---|---|---|---|
| | | | $AP^b$ | $AP^b_{50}$ | $AP^b_{75}$ | $AP^m$ | $AP^m_{50}$ | $AP^m_{75}$ | $AP^b$ | $AP^b_{50}$ | $AP^b_{75}$ | $AP^m$ | $AP^m_{50}$ | $AP^m_{75}$ |
| PVT-T (Wang et al., 2021) | 33 | 240 | 36.7 | 59.2 | 39.3 | 35.1 | 56.7 | 37.3 | 39.8 | 62.2 | 43.0 | 37.4 | 59.3 | 39.9 |
| MPViT-T (Lee et al., 2022) | 28 | 216 | 42.2 | 64.2 | 45.8 | 39.0 | 61.4 | 41.8 | 44.8 | 66.9 | 49.2 | 41.0 | 64.2 | 44.1 |
| RAVLT-T (Fan et al., 2025b) | 33 | 219 | 47.2 | 69.1 | 51.7 | 42.5 | 66.0 | 46.0 | 46.4 | 67.4 | 50.9 | 41.7 | 64.7 | 45.3 |
| MAViT-T (Fan et al., 2025a) | 33 | 219 | 47.5 | 69.0 | 52.3 | 42.8 | 66.3 | 46.3 | - | - | - | - | - | - |
| NaLaFormer-T | 33 | 226 | 47.6 | 69.5 | 52.4 | 43.0 | 66.7 | 46.5 | 46.7 | 67.4 | 51.3 | 42.0 | 65.0 | 45.7 |
| MPViT-S (Lee et al., 2022) | 43 | 268 | 46.4 | 68.6 | 51.2 | 42.4 | 65.6 | 45.7 | 48.4 | 70.5 | 52.6 | 43.9 | 67.6 | 47.5 |
| FL-Swin-T (Han et al., 2023) | 49 | 268 | 46.5 | 66.1 | 47.9 | 40.2 | 63.1 | 43.0 | 46.5 | 68.5 | 50.8 | 42.1 | 65.4 | 45.1 |
| VMamba-T (Liu et al., 2024) | 50 | 271 | 47.3 | 69.3 | 52.0 | 42.7 | 66.4 | 45.9 | 48.8 | - | - | 43.7 | - | - |
| MILA-T (Han et al., 2024b) | 44 | 255 | 46.8 | 69.5 | 51.5 | 42.1 | 66.4 | 45.0 | 48.8 | 71.0 | 53.6 | 43.8 | 68.0 | 46.8 |
| NaLaFormer-S | 44 | 272 | 49.5 | 71.2 | 54.3 | 44.2 | 68.1 | 47.8 | 49.7 | 70.5 | 54.7 | 44.3 | 68.0 | 48.0 |

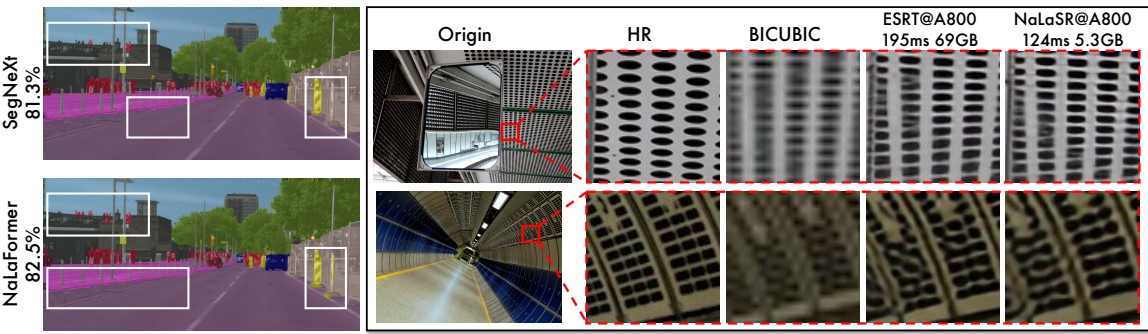

*Figure 5.* Visualizations illustrating NaLaFormer's semantic segmentation results on the CityScapes dataset (left) and NaLaSR and ESRT's super-resolution results on the Urban100 benchmark (right).

model with the ImageNet-1K pre-trained weight using mIoU as the evaluation metric, and train it following previous works (Han et al., 2023) on the *mmcv* (Contributors, 2018).

**Results.** As shown in Tab. 3, NaLaFormer achieves superior segmentation accuracy while maintaining favorable model complexity. On ADE20K, NaLaFormer-T and NaLaFormer-S achieve 46.9% and 48.5% mIoU respectively, bringing up to 4.7% and 2.0% improvements compared with models of similar scale. On Cityscape, NaLaFormer-T achieves 82.5% mIoU, consistently surpassing counterparts with comparable model sizes. As further illustrated in Fig. 5 left, the visualization on the Cityscapes dataset demonstrates that NaLaFormer captures sharper boundaries and richer structural details compared to SegNeXt (Guo et al., 2022b), highlighting its superiority in complex scenes. More visualizations are shown in Appendix A.8.

### 4.4. Super Resolution

**Settings.** We conduct the experiments on SR tasks following previous efficient SISR work, ESRT (Lu et al., 2022). All models are trained on DIV2K (Agustsson & Timofte, 2017), and are evaluated using PSNR and SSIM on reconstructed SR images. Meanwhile, we make statistics on both memory consumption and inference duration.

**Results.** To comprehensively evaluate our approach, we apply it to two distinct architectures: NaLa-ESRT based on the globally encoded ESRT (Lu et al., 2022), and NaLa-DCTLSA built upon the patch-wise DCTLSA (Zeng et al., 2023). As shown in Tab. 4, when compared with their respective base models, both NaLa variants achieve highly competitive PSNR and SSIM across all benchmarks, while

*Table 3.* Comparisons on the semantic segmentation tasks. The table on the left presents the results on the ADE20K dataset, while the table on the right shows the results on the Cityscapes dataset.

| METHOD | PARA | ADE20K | | CITYSCAPES | |
|---|---|---|---|---|---|
| | | mIoU | F | mIoU | F |
| VWFormer-B1 (Yan et al., 2024) | 14M | 44.0 | 13G | 80.4 | - |
| EfficientViT-B2 (Cai et al., 2023) | 15M | 45.9 | 9.1G | 82.1 | 74G |
| SegFormer-B1 (Xie et al., 2021) | 14M | 42.2 | 16G | 78.5 | 244G |
| SegNeXt-S (Xie et al., 2021) | 15M | 44.3 | 16G | 81.3 | 125G |
| NaLaFormer-T | 14M | 46.9 | 15G | 82.5 | 111G |
| VRWKV-S (Duan et al., 2025) | 29M | 47.2 | 46G | - | - |
| MambaOut-T (Yu & Wang, 2025) | 54M | 47.4 | - | - | - |
| VWFormer-B2 (Yan et al., 2024) | 27M | 48.1 | 47G | 81.7 | 415G |
| SegFormer-B2 (Xie et al., 2021) | 28M | 46.5 | 62G | 81.0 | 711G |
| NaLaFormer-S | 25M | 48.5 | 29G | 83.5 | 206G |

dramatically reducing inference latency and peak memory usage (e.g., saving up to 61.76% latency and 92.3% memory). Furthermore, Fig. 5 (right) presents a visual comparison on the $\times 4$ Urban100, where the cropped regions are enlarged for clarity. Despite the significant efficiency improvements, our models reconstruct sharper textures and more regular structures than the baselines. More results are provided in Tab. 20.

### 4.5. Diffusion Transformer

**Settings.** Following the settings of DiT (Peebles & Xie, 2023) and SiT (Ma et al., 2024), we further validate the effectiveness of NaLaFormer in the diffusion transformer framework. Specifically, we conduct experiments on the diffusion transformer S/2 architecture using ImageNet-1K (Deng et al., 2009). We replace the original attention module with NaLaFormer while keeping the remaining training configurations unchanged. All models are evaluated under the same conditions for a fair comparison.

*Table 4.* Quantitative comparison and efficiency analysis on benchmark datasets for image super-resolution. The best results are highlighted in **bold**. The "LAT" denotes the inference latency and "MEM" represents peak memory usage.

| MODEL | SCALE | SET5 | | SET14 | | BSD100 | | URBAN100 | |
|---|---|---|---|---|---|---|---|---|---|
| | | PSNR | SSIM | PSNR | SSIM | PSNR | SSIM | PSNR | SSIM |
| Bicubic | ×4 | 28.42 | 0.81 | 26.00 | 0.70 | 25.96 | 0.67 | 23.14 | 0.66 |
| LAPAR-B (Li et al., 2020) | ×4 | 31.94 | 0.89 | 28.46 | 0.78 | 27.52 | 0.73 | 25.85 | 0.79 |
| ECBSR (Zhang et al., 2021) | ×4 | 31.92 | 0.89 | 28.34 | 0.78 | 27.48 | 0.74 | 25.81 | 0.78 |
| ESRT (Lu et al., 2022) | ×4 | 32.01 | 0.89 | 28.44 | 0.77 | 27.48 | 0.73 | 25.85 | 0.78 |
| NaLa-ESRT | ×4 | 32.00 | 0.89 | 28.50 | 0.78 | 27.49 | 0.73 | 25.83 | 0.78 |
| SwinIR (Liang et al., 2021) | ×4 | 32.44 | 0.8976 | 28.77 | 0.7858 | 27.69 | 0.7406 | 26.47 | 0.7980 |
| DCTLSA (Zeng et al., 2023) | ×4 | 32.44 | 0.8973 | 28.77 | 0.7846 | 27.67 | 0.7386 | 26.41 | 0.7944 |
| NaLa-DCTLA | ×4 | **32.61** | **0.8992** | **28.91** | **0.7878** | **27.75** | **0.7414** | **26.74** | **0.8037** |
| Bicubic | ×3 | 30.39 | 0.87 | 27.55 | 0.77 | 27.21 | 0.74 | 24.46 | 0.73 |
| SRFBN-S (Li et al., 2019) | ×3 | 34.20 | 0.93 | 30.10 | 0.84 | 28.96 | 0.80 | 27.66 | 0.84 |
| LAPAR-B (Li et al., 2020) | ×3 | 34.20 | 0.93 | 30.17 | 0.84 | 29.03 | 0.80 | 27.85 | 0.85 |
| ESRN-V (Song et al., 2020) | ×3 | 34.23 | 0.93 | 30.27 | 0.84 | 29.03 | 0.80 | 27.95 | 0.85 |
| ESRT (Lu et al., 2022) | ×3 | 34.13 | 0.92 | 30.24 | 0.84 | 28.99 | 0.80 | **27.88** | 0.85 |
| NaLa-ESRT | ×3 | **34.21** | **0.93** | **30.24** | **0.84** | **29.00** | **0.80** | 27.87 | **0.85** |
| SwinIR (Liang et al., 2021) | ×3 | 34.62 | 0.9289 | 30.54 | 0.8463 | 29.20 | 0.8082 | 28.66 | 0.8624 |
| DCTLSA (Zeng et al., 2023) | ×3 | 34.76 | 0.9296 | 30.64 | 0.8474 | 29.27 | 0.8093 | 28.81 | 0.8674 |
| NaLa-DCTLA | ×3 | **34.81** | **0.9299** | **30.71** | **0.8486** | **29.32** | **0.8104** | **29.06** | **0.8693** |
| **Efficiency@RTX3090** | SCALE | LAT | MEM | LAT | MEM | LAT | MEM | LAT | MEM |
| NaLa-DCTLA | ×4 | **32.07ms** | **0.33GB** | **49.30ms** | **0.51GB** | **31.85ms** | **0.28GB** | **155.12ms** | **1.51GB** |
| SwinIR (Liang et al., 2021) | ×4 | 594.16ms | 2.67GB | 1297.00ms | 4.20GB | 884.83ms | 1.65GB | 4510.52ms | 12.44GB |
| - SAVE | ×4 | 94.60% | 87.64% | 96.20% | 87.86% | 96.40% | 83.03% | 96.56% | 87.86% |
| DCTLSA (Zeng et al., 2023) | ×4 | 57.70ms | 1.05GB | 109.04ms | 1.47GB | 72.06ms | 0.59GB | 349.86ms | 4.28GB |
| - SAVE | ×4 | 44.42% | 68.57% | 54.79% | 65.31% | 55.80% | 52.54% | 55.67% | 64.72% |
| NaLa-DCTLA | ×3 | **32.56ms** | **0.43GB** | **52.03ms** | **0.65GB** | **35.65ms** | **0.31GB** | **172.56ms** | **1.96GB** |
| SwinIR (Liang et al., 2021) | ×3 | 625.78ms | 2.65GB | 1330.60ms | 4.16GB | 869.78ms | 1.64GB | 4511.03ms | 12.34GB |
| - SAVE | ×3 | 94.80% | 83.77% | 96.09% | 84.38% | 95.90% | 81.10% | 96.17% | 84.12% |
| DCTLSA (Zeng et al., 2023) | ×3 | 57.90ms | 1.66GB | 113.04ms | 2.27GB | 76.74ms | 0.93GB | 355.81ms | 7.19GB |
| - SAVE | ×3 | 43.77% | 74.10% | 53.97% | 71.37% | 53.54% | 66.67% | 51.50% | 72.74% |

**Results.** As shown in Table 5, NaLaFormer consistently improves diffusion transformer performance. NaLaDiT outperforms DiT (Peebles & Xie, 2023) and DiG (Zhu et al., 2025), and NaLaSiT further achieves the best results among SiT-based variants (Ma et al., 2024; Pu et al., 2024), demonstrating the effectiveness of NaLaFormer in diffusion transformers. We visualize the generated images in Fig. 8.

*Table 5.* Diffusion transformer results. NaLaFormer achieves strong performance on both DiT and SiT.

| MODEL | FID ↓ | sFID ↓ | IS ↑ | PRECISION ↑ | RECALL ↑ |
|---|---|---|---|---|---|
| DiT (Peebles & Xie, 2023) | 68.40 | - | - | - | - |
| DiG (Zhu et al., 2025) | 62.06 | **11.77** | 22.81 | 0.39 | 0.56 |
| NaLaDiT | 61.64 | 12.40 | **23.24** | **0.40** | **0.58** |
| SiT (Ma et al., 2024) | 58.61 | 9.25 | 24.31 | 0.41 | 0.59 |
| EfficientSiT (Pu et al., 2024) | 53.57 | 9.01 | 27.26 | 0.43 | 0.61 |
| NaLaSiT | **53.08** | **8.94** | **27.63** | **0.43** | **0.62** |

### 4.6. Language Modeling

**Settings.** We train our model from scratch with parameter sizes of 340M and test it on common-sense reasoning tasks. Our method is integrated in DeltaNet (Yang et al., 2024b) and Gated DeltaNet (Yang et al., 2025a) by replacing $\text{SiLU}(\cdot)$ function with query norm-aware feature map.

**Results.** As shown in Tab. 6, baselines such as Deltanet (Yang et al., 2024b) and Gated DeltaNet (Yang et al., 2025a) demosntrate a consistent performance gain across various language reasoning tasks. By equipping with the proposed kernel functions, our model consistently outperform DeltaNet and Gated DeltaNet.

*Table 6.* Comparisons on common-sense reasoning tasks.

| MODEL | WIKI. ppl ↓ | LMB. ppl ↓ | PIQA acc ↑ | HELLA. $\text{acc}_n$ ↑ | WINO. acc ↑ | $\text{ARC}_e$ acc ↑ | $\text{ARC}_c$ $\text{acc}_n$ ↑ | AVG. |
|---|---|---|---|---|---|---|---|---|
| Transformer++ | 28.39 | 42.69 | 63.3 | 34.0 | 50.4 | 44.5 | 24.2 | 43.3 |
| RetNet | 32.33 | 49.19 | 63.5 | 33.5 | 52.5 | 44.5 | 23.4 | 43.5 |
| Mamba | 28.39 | 39.66 | 65.0 | 35.4 | 50.1 | 46.3 | 23.6 | 44.1 |
| GLA | 28.65 | 43.35 | 64.8 | 34.5 | 51.4 | 45.1 | 22.7 | 43.7 |
| DeltaNet | 29.08 | 50.87 | 63.6 | 33.6 | 51.7 | 46.0 | 23.0 | 43.6 |
| NaLa+DN | **27.82** | **49.77** | **64.9** | **34.3** | **52.7** | **46.5** | **23.1** | 44.3 +0.7 |
| Gated DeltaNet | 26.59 | **31.67** | **65.8** | 35.2 | 50.8 | **46.0** | 23.5 | 44.3 |
| NaLa+GDN | **25.89** | 32.32 | 65.6 | **36.2** | **53.2** | 45.4 | **23.8** | 44.8 +0.5 |

### 4.7. Efficiency Analysis

The efficiency comparison with methods of similar FLOPs on classification tasks, presented in Fig. 6, demonstrates that NaLaFormer matches or exceeds baseline accuracy with substantially reduced computation. Fig. 7 shows that, NaLaFormer attains competitive throughput across NLP tasks, outperforming softmax attention and surpassing other baselines. Furthermore, we evaluate the efficiency NaLaFormer on Long Range Arena (LRA) benchmarks, as shown in Tab. 7, NaLaFormer achieves strong performance, sustaining higher training throughput. Full results and details can be found in Appendix A.9.

*Table 7.* Results on LRA tasks compared with other efficient Transformer models. "THR" denotes throughput (in TGS) and "MEM" denotes peak memory usage (in MB).

| MODEL | ACC$_{avg}$ | THR$_{avg}$ | MEM$_{avg}$ |
|---|---|---|---|
| Softmax | 58.1 | 439.7 | 9004 |
| Kernelized | 56.6 | 528.5 | 9606 |
| Nystrom | 57.9 | 1007.7 | 2832 |
| Linformer | 55.1 | 918.8 | 1897 |
| Skyformer | 59.4 | 719.5 | 3985 |
| PolaFormer | 60.7 | 915.6 | 2047 |
| NaLaFormer | **61.2** | 827.7 | 2603 |

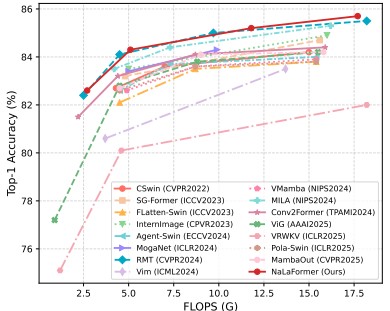

*Figure 6.* Efficiency analysis with Acc vs. FLOPs curves on the ImageNet-1K.

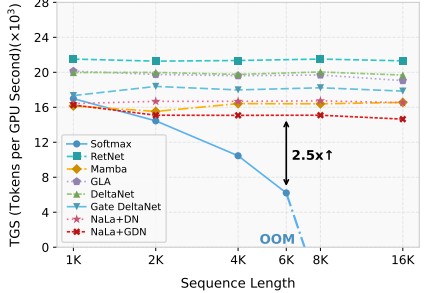

*Figure 7.* Comparison on training throughput of 340M models on a single A6000 GPU.

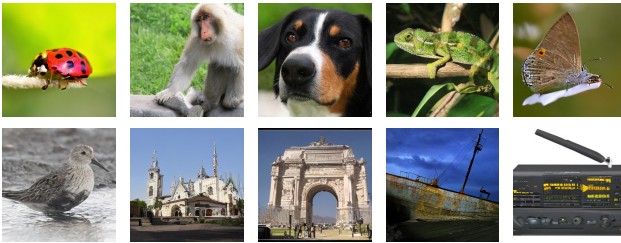

*Figure 8.* Visualizations of the NaLaDiT-XL/2 with $256 \times 256$.

### 4.8. Ablation Study

We conduct extensive ablations to validate NaLaFormer's norm-aware mechanism, structural robustness, and hyperparameters, with exhaustive details deferred to Appendix A.3. Crucially, to address concerns regarding training instability caused by pathological query norm growth at scale, we directly compare our method with QK-Norm. This comparison aims to verify whether our bounded norm-aware approach remains trainable and beneficial at larger scales.

**Components in Norm-aware Linear Attention.** We ablate NaLaFormer's core designs in Tab. 8. Compared to a baseline using $\mathrm{ReLU}(\cdot)$ and a constant power function (Row 1), introducing the cosine inhibit (Row 2) yields a 0.4% gain by preserving norm information from negative values. Upgrading to a norm-aware power function (Row 3) provides an additional 0.4% boost. This verifies that our method successfully recovers critical information typically lost during standard norm cancellation.

*Table 8.* Ablation of the Components in Norm-aware Linear Attention under FL-Swin-T setting.

| NON NEGATIVITY | SPIKY | NORM AWARE | COSINE DIR SIM | ACC. (%) |
|---|---|---|---|---|
| ✓ | ✓ | | | 82.1$_{-0.8}$ |
| ✓ | ✓ | | ✓ | 82.5$_{-0.4}$ |
| ✓ | ✓ | ✓ | ✓ | 82.9 |

**Comparison with Existing Linear Attention.** Adopting the FLatten-Transformer (Han et al., 2023) evaluation protocol under the Swin-T setting, we directly replace the attention module with NaLaFormer. As shown in Tab. 9, our method achieves consistent gains over all baseline models, surpassing both conventional linear attention variants and softmax attention while maintaining linear complexity.

*Table 9.* Comparison with other linear attentions.

| METHOD | PARAMS | FLOPs | ACC(%) |
|---|---|---|---|
| Swin-T (Liu et al., 2021) | 28M | 4.4G | 81.2 |
| HydraAttn (Bolya et al., 2022) | 29M | 4.5G | 80.7 |
| EfficientAttn (Shen et al., 2021) | 29M | 4.5G | 81.0 |
| LinearAngular (You et al., 2023) | 29M | 4.5G | 79.4 |
| EnhancedAttn (Cai et al., 2023) | 29M | 4.5G | 81.8 |
| FLattenAttn (Han et al., 2023) | 29M | 4.5G | 82.1 |
| AgentAttn (Han et al., 2024c) | 29M | 4.5G | 82.6 |
| PolaFormer (Meng et al., 2025) | 29M | 4.5G | 82.6 |
| InLine (Han et al., 2024a) | 30M | 4.5G | 82.4 |
| CA (Han et al., 2026) | 28M | 4.6G | 82.2 |
| NaLaFormer | 29M | 4.8G | **82.9** |

**Comparison with QK-Norm.** To verify that our gains do not stem solely from feature normalization, we compare NaLaFormer with QK-Norm (Tab. 15). Compared to the FLatten baseline, QK-Norm yields mixed results ($-0.04\%$ on Swin, $+0.15\%$ on PVT). In contrast, NaLaFormer consistently provides superior improvements ($+0.21\%$ on Swin, $+0.50\%$ on PVT). This validates that our bounded restoration of query-side norm information is actively beneficial, avoiding the performance compromises of simply discarding norm data.

**Ablation Study on $\tau$ and $\lambda$.** Finally, we conduct ablation studies on the hyperparameters $\tau$ and $\lambda$ across both image classification and document retrieval tasks from the LRA benchmark, with detailed results provided in Tab. 14.

## 5. Conclusion

In this work, we introduced NaLaFormer, a query norm-aware linear attention that restores the missing role of query norms and preserves non-negativity through cosine direction similarity. Our approach bridges the gap between softmax and linear attention by reducing the entropy in query norm awareness and avoid suppressing negative values. We validated the effectiveness of NaLaFormer across a wide range of vision tasks, including image classification, detection, segmentation, and super-resolution, as well as on language modeling and the Long Range Arena benchmark. The results consistently show that NaLaFormer achieves higher accuracy and better efficiency than existing linear attention models, offering a more practical balance between performance and efficiency.

## Acknowledgments

This research is partially supported by National Natural Science Foundation of China (Grant no. 62372132). The authors would also like to thank Huawei Ascend Cloud Ecological Development Project for providing high-performance Ascend 910 processors.

## Impact Statement

This paper advances the field of linear attention by, for the first time, revealing the relationship between query norms and attention entropy, and by providing principled guidelines for feature map design that future work can build upon. There are many potential societal consequences of our work, none of which we feel must be specifically highlighted here.

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

# A. Appendix

## A.1. Entropy Analysis

In this section, we use the Positive Sequence Entropy (PSE) (Meng et al., 2025) to connect the probability distribution with the sequence of query-key similarity (one row in the feature map). In the following derivation, we use the Positive Sequence Entropy (PSE) (Meng et al., 2025) to connect the softmax self-attention with $\mathrm{PSE}(\cdot)$. We investigate the probability distribution generated from one single query vector and a series of key vectors with PSE, analyzing how $\mathrm{PSE}(\mathbf{x})$ varying with query norm with softmax.

We first give the definition of PSE as following,

**Definition A.1** (Positive Sequence Entropy). Let a sequence $\mathbf{x} = (x_1, ..., x_N)$, in which $x_i \geq 0$, $i = 1, \ldots, N$, and $s = \sum_{i=1}^{N} x_i > 0$. The uncertainty of this positive sequence is defined by:

$$\mathrm{PSE}(\mathbf{x}) = -\sum_{i=1}^{N} \frac{x_i}{s} \log(\frac{x_i}{s}), \; s = \sum_{i=1}^{N} x_i. \tag{15}$$

Assuming $\mathbf{q}$ is a directional fixed vector with norm $c$, *i.e.,* $\mathbf{q_t} = c_t \cdot d(\mathbf{q_t})$, we only consider the relation between PSE and $c_t$. Then, we have the following two theorems:

**Theorem A.2** (Query Norm-aware Entropy Reduction in Softmax Attention). *Given that* $\mathbf{x}_i = \mathbf{q}\mathbf{k}_i^\top$ *be a positive sequence and let* $\Phi : (-\infty, +\infty) \mapsto [0, +\infty)$ *be a spiky function serving to reduce the PSE through mapping each* $x_i$. *In the case* $\Phi(\cdot) = \exp(\cdot)$, *existing a constant value* $c_0$ *satisfying: For* $c > c_0$, *we have*

$$\mathrm{PSE}(\Phi((c\mathbf{q})\mathbf{k}^\top)) = \mathrm{PSE}(\Phi(c\mathbf{x})) < \mathrm{PSE}(\Phi(\mathbf{x})).$$

*Proof.* Without loss of generality, we assume the sum of the positive sequence have, $\sum_{i=1}^{N} \Phi(x_i) = 1$, and directly set query norm as a scaler $c \in \mathbb{R}$. Then, the PSE of $\mathbf{x}$ degraded into Shannon entropy as,

$$\mathrm{PSE}(\Phi(\mathbf{x})) = \mathrm{H}(\Phi(\mathbf{x}))$$
$$= -\sum_{i=1}^{N} \Phi(x_i) \log(\Phi(x_i))$$

When query norm varies, the PSE is,

$$S_c = \sum_{i=1}^{N} \Phi(cx_i) = \sum_{i=1}^{N} \Phi(x_i)^c = |||\Phi(\mathbf{x})||_c^c$$

$$\text{PSE}(\Phi(\mathbf{cx})) = \text{PSE}(\Phi(\mathbf{x})^c) = \log(S_c) - \sum_{i=1}^{N} \frac{\Phi(x_i)^c}{S_c} \log(\Phi(x_i)^c)$$

$$= c\log(||\Phi(\mathbf{x})||_c) - c\sum_{i=1}^{N} \frac{\Phi(x_i)^c}{||\Phi(\mathbf{x})||_c} \log(\Phi(x_i))$$

Due to the definition of $L_p$ norm, $\lim_{p \to \infty} ||x||_p = x_{max}$, and $x_i \in [0,1]$, we have $||x||_p <= x_{max}$ for $p > 1$. Therefore,

$$c\log(||\Phi(\mathbf{x})||_c) - c\sum_{i=1}^{N} \frac{\Phi(x_i)^c}{||\Phi(\mathbf{x})||_c} \log(\Phi(x_i)) \leq c\log(\Phi(x_{max})) - c\sum_{i=1}^{N} (\frac{\Phi(x_i)}{\Phi(x_{max})})^c \log(\Phi(x_i))$$

When $c \to +\infty$, $(\frac{\Phi(x_i)}{\Phi(x_{max})})^c \to 0$ for all $x_i \neq x_{max}$:

$$\lim_{c \to +\infty} c\log(\Phi(x_{max})) - c\sum_{i=1}^{N} (\frac{\Phi(x_i)}{\Phi(x_{max})})^c \log(\Phi(x_i))$$

$$= c\log(\Phi(x_{max})) - c\log(\Phi(x_{max})) = 0$$

Therefore, because PSE is positive, there exists $c_0$, for all $c > c_0$, $\text{PSE}(\Phi(c\mathbf{x})) < \text{PSE}(\Phi(\mathbf{x}))$  $\square$

Consequently, the Theorem A.2 proves the theorem softmax attention is query norm aware with a dynamic control on entropy reduction.  ∎

Similar to linear attention, we continue with the case in previous linear attention with feature maps, and prove that the PSE of existing linear attentions is query norm-unaware.

---

**Theorem A.3** (Query Norm-unaware of Entropy in Linear Attention). *Given that* $\mathbf{x} = (x_1, \ldots, x_N)$, $\mathbf{x}_c = (cx_1, \ldots, cx_N)$, *are positive sequences, where* $c > 0$ *denotes the ratio of the query norm, and* $\phi(\cdot)$ *is a element-wise feature map satisfying* $c_1\phi(\mathbf{q}) \leq \phi(c\mathbf{q}) \leq c_2\phi(\mathbf{q})$. *Then, we have*

$$|\text{PSE}(\Phi(\mathbf{x}_c)) - \text{PSE}(\Phi(\mathbf{x}))| \leq \log(\frac{c_2}{c_1}) + \frac{c_2 - c_1}{c_1} \text{PSE}(\Phi(\mathbf{x})).$$

---

*Proof.* For most of the linear attentions, such as vanilla linear attention (Katharopoulos et al., 2020), FLatten (Han et al., 2023), Efficientvit (Cai et al., 2023) and PolaFormer (Meng et al., 2025), they all have $c_1\phi(\mathbf{q}) \leq \phi(c\mathbf{q}) \leq c_2\phi(\mathbf{q})$, and for ReLU$(\cdot)$ feature map, $c_1 = c_2$. Therefore, we have the following derivations:

If the feature map is a linear transformation, *i.e.*, $\Phi(x_m) = \phi(\mathbf{q})\phi(\mathbf{k}_m)^\top$ and $\Phi(c\mathbf{x}) = c\Phi(\mathbf{x})$, such as $\text{ReLU}(\cdot)$, we have,

$$S = \sum_{m=1}^{N} \Phi(x_m)$$

$$\text{PSE}_{\text{linear}}(\mathbf{x}) = \log(S) - \sum_{i=1}^{N} \frac{\Phi(x_i)}{S} \log(\Phi(x_i))$$

$$\text{PSE}_{\text{linear}}(c\mathbf{x}) = \log(c) + \log(S) - \sum_{i=1}^{N} \frac{c\,\Phi(x_i)}{c \cdot S} (\log(\Phi(x_i)) + \log(c))$$

$$= \log(c) + \log(S) - \sum_{i=1}^{N} \frac{c\,\Phi(x_i)}{c \cdot S} (\log(\Phi(x_i))) - \sum_{i=1}^{N} \frac{c\,\Phi(x_i)}{c \cdot S} \log(c)$$

$$= \log(S) - \sum_{i=1}^{N} \frac{c\,\Phi(x_i)}{c \cdot S} (\log(\Phi(x_i))) + \log(c) - \sum_{i=1}^{N} \frac{c\,\Phi(x_i)}{c \cdot S} \log(c)$$

$$= \text{PSE}_{\text{linear}}(\mathbf{x}).$$

For a linear attention with nonlinear feature map, such as Log-Normal Attention (Nahshan et al., 2024), FLatten (Han et al., 2023), Efficientvit (Cai et al., 2023) and PoLaFormer (Meng et al., 2025), they all have $c_1\phi(\mathbf{q}) \le \phi(c\mathbf{q}) \le c_2\phi(\mathbf{q})$ (and for $\text{ReLU}(\cdot)$ feature map, $c_1 = c_2$), thus, we have:

For clearity, we suppose the original positive sequence is normalized, *i.e.*, $\sum_{m=1}^{N} \Phi(x_m) = \sum_{m=1}^{N} \phi(\mathbf{q})\phi(\mathbf{k}_m)^\top = 1$, then, under the assumption $c_1\phi(\mathbf{q}) \le \phi(c\mathbf{q}) \le c_2\phi(\mathbf{q})$, we have

$$c_1\Phi(x_m) \le \Phi_c(x_m) := \phi(c\mathbf{q})\phi(\mathbf{k}^\top) \le c_2\Phi(x_m) \tag{16}$$

$$S_c = \sum_{i=1}^{N} \Phi_c(x_i) \tag{17}$$

$$c_1 S \le S_c \le c_2 S \tag{18}$$

$$S = 1 \tag{19}$$

$$PSE(\Phi(\mathbf{x})) = \log(S) - \sum_{i=1}^{N} \frac{\Phi(x_i)}{S} \log(\Phi(x_i)) \tag{20}$$

$$= \sum_{i=1}^{N} \Phi(x_i) \log(\Phi(x_i)) \quad (S{=}1) \tag{21}$$

$$PSE(\Phi(\mathbf{x}_c)) = \log(S_c) - \sum_{i=1}^{N} \frac{\Phi_c(x_i)}{S_c} \log(\Phi_c(x_i)) \tag{22}$$

$$\le \log(c_2) - \sum_{i=1}^{N} \frac{\Phi_c(x_i)}{S_c} \log(\Phi_c(x_i)). \tag{23}$$

Since $S_c > 0$ and $\Phi_c(x_m) > 0$, we have

$$\log(c_2) - \sum_{i=1}^{N} \frac{\Phi_c(x_i)}{S_c} \log(\Phi_c(x_i)) \tag{24}$$

$$\le \log(c_2) - \sum_{i=1}^{N} \frac{\Phi_c(x_i)}{S_c} (\log(\Phi(x_i)) + \log(c_1)) \tag{25}$$

$$= \log(\frac{c_2}{c_1}) - \sum_{i=1}^{N} \frac{\Phi_c(x_i)}{S_c} \log(\Phi(x_i)). \tag{26}$$

Due to $\sum_{m=1}^{N} \Phi(x_m) = 1, \Phi(x_m) \geq 0$, we have $\Phi(x_m) \leq 1, \log(\Phi(x_m)) \leq 0$ then

$$\log(\frac{c_2}{c_1}) - \sum_{i=1}^{N} \frac{\Phi_c(x_i)}{S_c} \log(\Phi(x_i)) \tag{27}$$

$$\leq \log(\frac{c_2}{c_1}) - \sum_{i=1}^{N} \frac{c_2 \Phi(x_i)}{c_1 S} \log(\Phi(x_i)) \tag{28}$$

$$= \log(\frac{c_2}{c_1}) - \frac{c_2}{c_1} \sum_{i=1}^{N} \Phi(x_i) \log(\Phi(x_i)) \tag{29}$$

$$= \log(\frac{c_2}{c_1}) + \frac{c_2}{c_1} PSE(\Phi(\mathbf{x})). \tag{30}$$

Similar with the derivations above, we have the lower bound of $PSE(\mathbf{x}_c)$,

$$PSE(\Phi(\mathbf{x}_c)) \geq \log(\frac{c_1}{c_2}) + \frac{c1}{c2} PSE(\Phi(\mathbf{x})). \tag{31}$$

Therefore,

$$\log(\frac{c_1}{c_2}) + (\frac{c1}{c2} - 1)PSE(\Phi(\mathbf{x})) \leq PSE(\Phi(\mathbf{x}_c)) - PSE(\Phi(\mathbf{x})) \leq \log(\frac{c_2}{c_1}) + (\frac{c_2}{c_1} - 1)PSE(\Phi(\mathbf{x})). \tag{32}$$

Since $c_2 > c_1 > 0$, we have $\log(\frac{c2}{c1}) > 0$ and ,

$$|PSE(\Phi(\mathbf{x}_c)) - PSE(\Phi(\mathbf{x}))| \leq \log(\frac{c2}{c_1}) + \frac{c_2 - c_1}{c_1} PSE(\Phi(\mathbf{x})) \tag{33}$$

Here, both $c_1$ and $c_2$ vary with $c$.

For example, as the feature map of Linear Log-Normal Attention (Nahshan et al., 2024), $\phi(\mathbf{q}) = \exp(\mathbf{q})$, we have

$$\exp(\min_{d}(\mathbf{q}_d)) \cdot \Phi(x_m) \leq \Phi_c(x_m) \leq \exp(\max_{d}(\mathbf{q}_d)) \cdot \Phi(x_m), \tag{34}$$

$$|PSE(\Phi(\mathbf{x}_c)) - PSE(\Phi(\mathbf{x}))| \leq \log(\frac{c2}{c_1}) + \frac{c_2 - c_1}{c_1} PSE(\Phi(\mathbf{x})) \tag{35}$$

$$= \mathbf{q}_{max} - \mathbf{q}_{min} + (\exp(\mathbf{q}_{max} - \mathbf{q}_{min}) - 1)PSE(\Phi(\mathbf{x})) \tag{36}$$

From the equation above and the properties of exp function, it can be seen that when the query norm changes, the $PSE(\Phi(\mathbf{x_c}))$ of linear attention only fluctuates around $\Phi(PSE(\mathbf{x}))$, showing no negative correlation with the query norm. □

According to the derivations about Eq. (36), it is evident that the error, $|PSE(\Phi(\mathbf{x}_c)) - PSE(\Phi(\mathbf{x}))|$, is controlled by the feature map. Only when the steepness (*i.e.*, the second derivative) of the feature map function varies with the query norm can the query norm directly affect the variation of entropy in linear attention. Considering the Lemma 2 in PolaFormer (Meng et al., 2025), the composite function of the element-wise feature map with first and second derivative is concave, thus the feature map we proposed, power function with the exponent greater than 1 as well as changing with query norm can compensate for the property in softmax attention where the query norm influences PSE.

## A.2. Experiment Settings

**Implementation Details.** Building upon the framework illustrated in Fig. 2, we construct a hierarchical vision backbone NaLaFormer. Consistent with established works (Fan et al., 2025b; 2024; Liu et al., 2021), we develop a set of NaLaFormer backbones, each with varying configurations of block count and channel dimensions across their respective stages whose the ratio of MLP is set to 3.5. The architecture details are illustrated in the Tab. 10.

*Table 10.* Architecture details of NaLaFormer.

| Model | Blocks | Channels | Heads |
|---|---|---|---|
| NaLaFormer-XT | [2, 2, 4, 2] | [32, 64, 192, 384] | [1, 2, 6, 12] |
| NaLaFormer-T | [2, 2, 6, 2] | [64, 128, 256, 512] | [1, 2, 4, 8] |
| NaLaFormer-S | [3, 5, 9, 3] | [64, 128, 320, 512] | [1, 2, 5, 8] |
| NaLaFormer-B | [4, 6, 12, 6] | [96, 192, 384, 512] | [1, 2, 6, 8] |
| NaLaFormer-L | [4, 7, 19, 8] | [96, 192, 448, 640] | [1, 2, 7, 10] |

**Image Classification.** In this task, we train all of our models with AdamW optimizer for 320 epochs, including 20 epochs for linear warm-up. The basic learning rate is set to 0.001 for 128 micro batchsize and 1024 global batchsize. The training framework is developed on the top of the official DeiT implementation. Additionally, we use CPE (Chu et al., 2023) to serve as the positional encoding. When mapping each $d(\mathbf{x})$, we set $f(x) = \frac{\pi}{4}\tanh(x)$ to make the cosine function only inhibits the directions with opposite signals.

**Object Detection and Segmentation.** We further conducted comprehensive experiments on the object detection task using the COCO dataset (Lin et al., 2014), which contains 118K training images and 5K validation images annotated with 80 object categories. We use our model as backbone with pretrained weights on ImageNet-1K. We conduct the experiments following the *mmcv-detection* (Contributors, 2018) project. The model are trained under both $1\times$ (12 epochs) and $3\times$ (36 epochs). We use the AdamW optimizer with 0.0001 learning rate, 0.0001 weight decay and "step" policy.

**Semantic Segmentation.** We conduct the semantic segmentation of ADE20K dataset (Zhou et al., 2019). This widely adopted dataset comprises 25,000 densely annotated images depicting complex real-world environments with rich contextual interactions between objects and their spatial configurations. We employ the pretrained NaLaFormer models on two representative segmentation models, SemanticFPN and UperNet. The experiment is conducted based on *mmcv-segmentation* (Contributors, 2018). All models are trained using AdamW optimizer with 0.0001 learning rate and 0.001 weight decay.

**Super Resolution.** To both evaluate the accuracy of our method under super resolution and highlight the computational efficiency advantage that its linear complexity offers in super-resolution, we make the experiments following previous efficient SISR work, ESRT (Lu et al., 2022). We use DIV2K (Agustsson & Timofte, 2017) as the training dataset, and for evaluation, we use four benchmark datasets, including Set5, Set14, BSD100 and Urban100 as used in ESRT, utilizing both PSNR and SSIM to evaluate the performance of the reconstructed SR images. Meanwhile, we conducted statistics on both memory consumption and inference duration.

**Diffusion Transformer.** We conduct diffusion model experiments on ImageNet-1K (Deng et al., 2009) to evaluate the effectiveness of NaLaFormer in generative modeling. Following DiT (Peebles & Xie, 2023) and SiT (Ma et al., 2024), we adopt the diffusion transformer S/2 architecture as the main experimental setting. We integrate NaLaFormer into both DiT and SiT by replacing the original attention module while keeping the remaining architectures and training configurations unchanged for fair comparisons. All models are trained and evaluated under the same protocol as prior works, and the generative quality is measured using standard metrics including FID, sFID, and IS.

**Language Modeling.** We compare NaLaFormer with several baseline models, including Transformer++ (Touvron et al., 2023), Gated Linear Attention (Yang et al., 2024a), RetNet (Sun et al., 2023), Mamba (Gu & Dao, 2023), DeltaNet (Yang et al., 2024b) and Gated DeltaNet (Yang et al., 2025a). Each model is pretrained on the subset of the SlimPajama dataset. We train our model from scratch with parameter sizes of 340M on 15B tokens with a batch size of 0.5M tokens and test it on common-sense reasoning tasks, which includes WikiText (Merity et al., 2017), LAMBADA (Paperno et al., 2016), ARC-easy (Clark et al., 2018), ARC-challenge (Clark et al., 2018), HellaSwag (Zellers et al., 2019), PiQA (Bisk et al., 2020) and WinoGrande (Sakaguchi et al., 2020). All downstream tasks are conducted based on *lm-evaluation-harness*. We test throughput of the baseline models on a single A6000 GPU.

## A.3. Ablation Study

**Impact of Components in Norm-aware Linear Attention.** We evaluate the effectiveness of each component in NaLaFormer. In row 1, we keep non-negativity and spikiness with $\mathrm{ReLU}(\cdot)$ and a constant power function. In row 2, we utilize the cosine inhibit to additionally preserve the norm. In row 3, we replace the constant power with a norm aware power. As shown in Table 11, it is important to note that the norm awareness yields a 0.4% improvement in row 2 and row 3, indicating that norm-aware spikiness effectively capture the lost information due to the norm cancellation. We examine the impact of norm consistency with cosine inhibit in row 1 and row 2 by only preserving negative values, with our cosine inhibit, the information in negative values improves the performance 0.4%.

**Impact of Components in Vision Model with NaLaFormer.** The ImageNet classification experiments are conducted on top of the current sota method, RALA (Fan et al., 2025b). To ensure a fair comparison with the RALA baseline, we follow its model design. In order to verify that the performance gains of our method indeed stem from the advanced linear-attention mechanism, we further include the following ablation studies under the XT-size settings: blocks [2, 2, 4, 2], channels [32, 64, 192, 384], and heads [1, 2, 6, 12]. The results indicate that the influence of these components on model performance is limited, thereby further demonstrating the superiority of our norm-aware linear attention. The results are shown in Table. 13

**Comparison with other Linear Attention.** To ensure fair comparison with existing linear attention approaches, we adopt the evaluation protocol from FLatten-Transformer (Han et al., 2023) with Swin-T setting by only replacing the attention mechanism to our Norm-aware linear attention. As shown in Table 12, NaLaFormer achieves consistent performance gains across all baseline models, surpassing both conventional linear attention variants and softmax attention, while maintaining linear complexity.

**Comparison with QK-Norm.** To evaluate whether the performance gains stem solely from feature normalization, we conduct a direct comparison with QK-Norm under identical backbones and training configurations. As summarized in Tab. 15, on the Swin backbone, QK-Norm, FLatten, and NaLaFormer achieve Top-1 accuracies of 80.58%, 80.62%, and 80.83%, respectively. On the PVT backbone, the corresponding accuracies are 75.06%, 74.91%, and 75.41%. Compared to the baseline FLatten, QK-Norm exhibits mixed behavior, slightly decreasing performance on Swin by 0.04% while yielding a 0.15% improvement on PVT. In contrast, NaLaFormer consistently outperforms FLatten across both backbones (+0.21% on Swin and +0.50% on PVT). These empirical results demonstrate that the improvements of NaLaFormer cannot be attributed to normalization effects alone. Instead, they validate the effectiveness of our proposed bounded restoration of critical query-side norm information.

**Ablation Study in $\tau$ and $\lambda$.** We conduct the ablation study on both image classification (CV) and document retrieval (NLP) from LRA benchmark. The results are shown in Table 14.

*Table 11.* Ablation on the FL-Swin-T setting.

| NON NEGATIVITY | SPIKY | NORM AWARE | COSINE DIR SIM | ACC. (%) |
|---|---|---|---|---|
| ✓ | ✓ | | | $82.1_{-0.8}$ |
| ✓ | ✓ | | ✓ | $82.5_{-0.4}$ |
| ✓ | ✓ | ✓ | ✓ | 82.9 |

*Table 13.* Ablation studies of vision models with NaLaFormer-XT.

| w.o. | RoPE | CPE | Layerscales | Swish | Ours-XT |
|---|---|---|---|---|---|
| ACC | 79.1% | 78.9% | 79.1% | 78.7% | 79.1% |

*Table 12.* Comparison with other linear attention models on the Swin-T setting.

| METHOD | PARAMS | FLOPs | ACC(%) |
|---|---|---|---|
| Swin-T (Liu et al., 2021) | 28M | 4.4G | 81.2 |
| Hydra Attn (Bolya et al., 2022) | 29M | 4.5G | 80.7 |
| Efficient Attn (Shen et al., 2021) | 29M | 4.5G | 81.0 |
| Linear Angular (You et al., 2023) | 29M | 4.5G | 79.4 |
| Enhanced Attn (Cai et al., 2023) | 29M | 4.5G | 81.8 |
| FLatten Attn (Han et al., 2023) | 29M | 4.5G | 82.1 |
| Agent Attn (Han et al., 2024c) | 29M | 4.5G | 82.6 |
| InLine Attn (Han et al., 2024a) | 30M | 4.5G | 82.4 |
| PolaFormer (Meng et al., 2025) | 29M | 4.5G | **82.6** |
| NaLaFormer | 29M | 4.8G | **82.9** |

*Table 14.* Ablation studies in $\tau$ and $\lambda$.

| $\lambda$ | $\tau$ | RETRIEVAL (NLP) | IMAGE (CV) |
|---|---|---|---|
| 3 | 0.5 | 80.42 | 44.54 |
| 5 | 0.5 | 80.17 | 41.91 |
| 7 | 0.5 | 80.13 | 40.73 |
| 3 | 1 | 80.13 | 41.80 |
| 3 | 2 | 80.35 | 42.12 |

*Table 15.* Comparison with QK-Norm and FLatten baselines on Swin and PVT backbones. Top-1 accuracy (%) is reported.

| Method | Swin | PVT |
|--------|------|-----|
| QK-Norm | 80.58 | 75.06 |
| FLatten | 80.62 | 74.91 |
| **Ours** | **80.83** | **75.41** |

## A.4. Limitations

While this work validates the efficacy of NaLaFormer across diverse vision-language tasks, we anticipate that its linear self-attention architecture holds significant potential for other cross-modality applications, such as text-to-image and text-to-video generation. However, direct evaluation in such contexts presents considerable challenges, primarily due to the substantial computational complexity associated with training diffusion models from scratch. Future efforts will actively pursue these promising directions and develop novel strategies to accelerate training efficiency, thereby enabling scalable deployment of NaLaFormer in complex generative modeling tasks.

## A.5. Related Work

**Vision Transformer.** The success of the Transformer architecture (Vaswani et al., 2017) in natural language processing (NLP), particularly its self-attention mechanism for modeling long-range dependencies, has catalyzed its adoption in computer vision (CV). The vision transformer (Dosovitskiy et al., 2021) marked a paradigm shift by discarding convolutions entirely. The vision transformer partitions images into patches, linearly embeds these patches into sequential tokens, and processes them through a pure Transformer encoder. Nevertheless, the quadratic computational complexity inherent in self-attention mechanisms incurs substantial computational overhead, rendering ViT training computationally intensive. Existing researches have proposed multiple strategies to enhance ViT's efficiency. For instance, DeiT (Touvron et al., 2021) achieves data-efficient training through knowledge distillation, whereas the Swin Transformer (Liu et al., 2021) employs shifted window mechanisms to balance local feature extraction with global context modeling while maintaining linear complexity. These advancements have established Transformer-based architectures as foundational frameworks for visual tasks, effectively bridging the methodological divide between NLP-oriented architectures and CV's inherent geometric constraints. However, these improvements primarily address architectural adaptations rather than resolving the fundamental limitations of softmax-based attention mechanisms, thereby retaining significant training costs. Recent studies have explored alternative paradigms for visual representation learning to mitigate these constraints. Building on sequential image processing principles, several approaches employ state space models (SSMs) for patch encoding. Notably, VMamba (Liu et al., 2024; Huang et al., 2024) leverages SSM-based encoding through raster-scan ordering to extract hierarchical features while preserving the theoretical guarantee of linear computational complexity inherent to SSMs. In addition, VHeat (Wang et al., 2024) reconceptualizes image understanding through thermodynamic simulations, modeling image patches as heat sources, and analyzing thermal conduction processes, reducing the complexity to $\mathcal{O}(N^{1.5})$ through discrete cosine transforms (DCT) and inverse DCT operations.

**Linear Attention.** Linear attention employs kernel-based similarity approximation to circumvent the $\exp(\mathbf{q}\mathbf{k}^\top)$ in standard softmax attention. The foundational work (Katharopoulos et al., 2020) introduces a linear separable kernel $\phi(\cdot)$ as an alternative to the $\exp$ operator, exploiting the associative property of matrix multiplication to reduce computational complexity from $\mathcal{O}(N^2)$ to $\mathcal{O}(N)$. Subsequent variants adopt this Softmax-free paradigm with diverse kernel functions, including ReLU (Han et al., 2023; Cai et al., 2023), 1+ELU (Katharopoulos et al., 2020) and SiLU (Yang et al., 2024b; MiniMax et al., 2025). Furthermore, to enhance position awareness, Cosformer (Qin et al., 2022) integrates ReLU with Ptolemy's theorem, incorporating locality inductive biases through feature map re-weighting while empirically enforcing non-negativity constraints. Beyond kernel design, recent studies focus on preserving the spikiness property inherent in softmax attention. Hedgehog (Zhang et al., 2024) and MB-TaylorFormer (Qiu et al., 2023) employ series expansions to approximate the $\exp$ function, while FLatten Transformer (Han et al., 2023) and PolaFormer (Meng et al., 2025) utilize power functions to sharpen attention distributions. Notably, lightning attention (Qin et al., 2024) combines SiLU kernels with a gate mechanism, achieving scalability up to 456B parameters (MiniMax et al., 2025). Inline (Han et al., 2024a) provides an important insight by proving that the softmax function is injective in most cases, whereas linear attention is not. By modifying the normalization scheme, it restores the injectivity of linear attention. In addition, Inline introduces a local-attention residual (a convolution module) to enhance local bias, thereby compensating for softmax's strong capability in

modeling local patterns. This work highlights the importance of injectivity in linear attention and uses vectors with identical norms but different directions as counterexamples to address this limitation. However, Inline overlooks the relationship between attention distribution uncertainty and the query/key norms—an essential property of the softmax function. MALA (Fan et al., 2025a) notices the neglect of norms, its simple non-negative constraint on the feature map causes negative values to be ignored, thereby leading to information loss during inner product computation. In autoregressive architectures, linear attention enables RNNs parallelization through unidirectional encoding. Gated Linear Attention (Yang et al., 2024a) enhances this capability via data-dependent gating on $\mathbf{K}^\top \mathbf{V}$ hidden states, demonstrating superior performance in length generalization and recall-intensive tasks. Existing kernel functions exhibit performance degradation compared to standard softmax attention. MetaLA (Chou et al., 2024) constructs a lightweight recurrent-form linear attention by defining the optimal linear approximation conditions of the softmax attention map, however, when applied to encoder architectures, such as ViT models or bidirectional attention, the model performance becomes sensitive to the scanning order, making it less suitable for vision tasks. Existing kernel-based linear attention mechanisms generally suffer from performance degradation compared to standard softmax attention.

### A.6. Future Work

This work reveals the fundamental relationship between query norms and attention entropy and introduces a norm-aware linear attention mechanism that restores this property. In future research, we will further explore the interaction between our method and different positional encoding schemes (Su et al., 2024; Yang et al., 2025b; Chu et al., 2023; Xie et al., 2025), as the observed performance variations mainly stem from how positional encodings capture absolute or relative positional information. We also plan to investigate a broader family of Injection schemes beyond the current power-function design, including alternative spiky mappings such as exponential forms (Nahshan et al., 2024). In addition, while our method is primarily developed for Vision Transformers, extending norm-aware feature maps to decoder-only Transformer architectures remains a promising direction. Building on this, we also aim to integrate our efficient backbone into Vision-Language Models (VLMs) to fundamentally resolve the quadratic computational bottleneck combining with current representation-level compression methods (Zhuang et al., 2026; Liang et al., 2026; 2025). Furthermore, we plan to validate the generalizability and scalability of our framework through larger-scale experiments, evaluating its performance across more extensive datasets and significantly larger model parameters. Finally, for practical deployment in resource-constrained settings, techniques such as quantization will be explored to further improve efficiency.

### A.7. Implement Discussion

Our method primarily operates on the feature map before the linear attention computation. Following the norm-aware spiky mapping and the cosine-based transformation of the model inputs, attention is computed using the standard linear attention formulation, which naturally enables integration with Triton operator optimizations developed for FLA. However, since FLA is primarily designed for causal attention, for vision tasks we instead use fbi_la, a library also maintained by the FLA (Yang & Zhang, 2024) team, which provides optimized implementations for bidirectional linear attention through triton. Given that our algorithm consists purely of matrix multiplications, it can directly leverage the Triton implementation of simple linear attention, shown in Tab. 16. Furthermore, the pure matrix-multiplication structure is highly compatible with quantization techniques, leaving substantial room for further optimization, thus our method is more friendly with quantization method. Building on our current approach, we plan to explore hardware-level optimizations in future work.

*Table 16.* Comparison on implement methods

| IMPLEMENT | TIME | MAX MEMORY |
|---|---|---|
| PyTorch | 0.2617 | 8110 |
| FLA | 0.2357 (save 10%) | 7753 (save 5%) |

## A.8. More Visualizations

More visualizations are provided in this section.

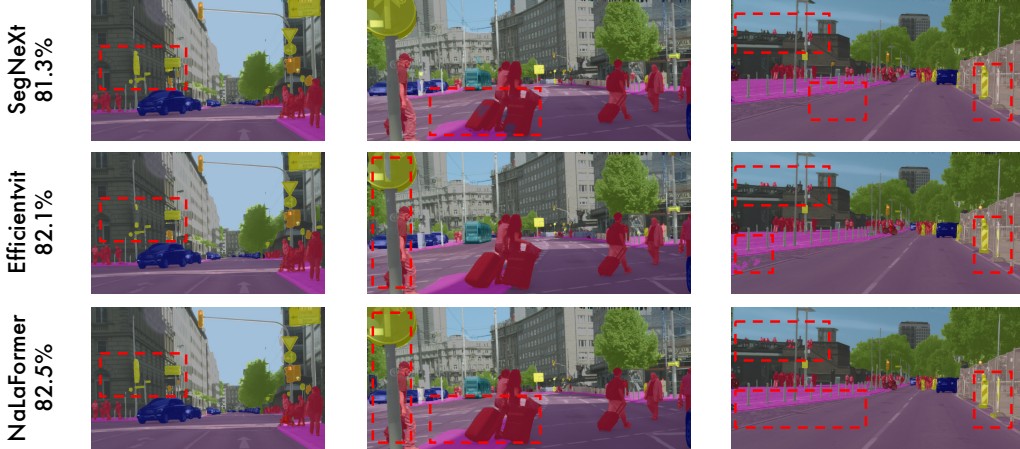

*Figure 9.* Comparison of the visualizations among different models.

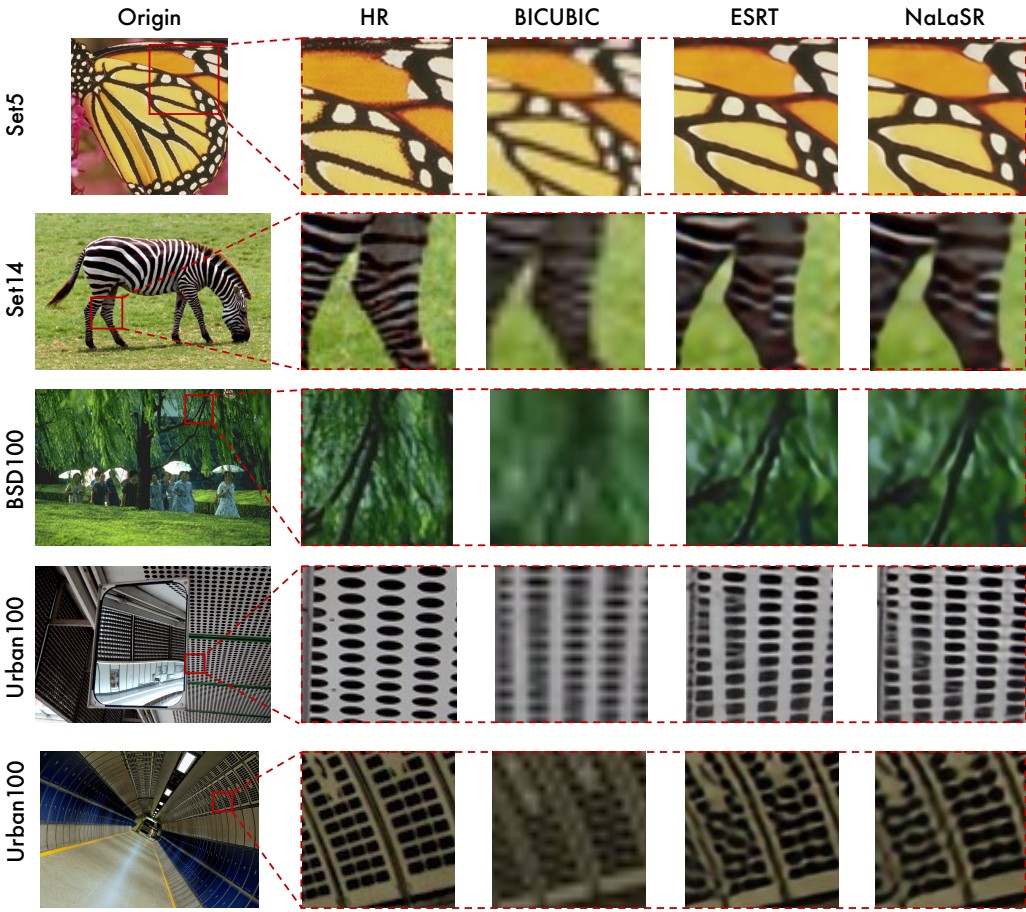

*Figure 10.* Visualizations of the NaLaSR in four different benchmarks comparing with ESRT under ×4 scale.

## A.9. Full Tables

Full tables of the experimental results are shown in this section.

*Table 17.* Object detection and instance segmentation results on the COCO dataset using RetinaNet with 1 × schedule.

| METHOD | TYPE | RETINANET 1× | | | | | |
|---|---|---|---|---|---|---|---|
| | | $AP^b$ | $AP^b_{50}$ | $AP^b_{75}$ | $AP^b_S$ | $AP^b_M$ | $AP^b_L$ |
| PVTv2-b1 (Wang et al., 2022) | Trans | 41.2 | 61.9 | 43.9 | 25.4 | 44.5 | 54.3 |
| SBCFormer-L (Lu et al., 2024b) | Trans | 41.1 | 62.3 | 43.3 | 24.7 | 44.3 | 56.0 |
| MPViT-XS (Lee et al., 2022) | Trans | 45.9 | 67.4 | 49.4 | 28.5 | 50.1 | 60.8 |
| SOFT + +-T (Lu et al., 2024a) | Linear | 41.9 | 62.7 | 44.7 | 27.8 | 45.4 | 55.6 |
| Pola-PVT-T (Meng et al., 2025) | Linear | 40.0 | 60.7 | 42.3 | 25.0 | 43.6 | 52.9 |
| NaLaFormer-T | Linear | **46.2** | **67.9** | **49.5** | **29.9** | **50.4** | **61.6** |
| MPViT-S (Lee et al., 2022) | Trans | 45.7 | 57.3 | 48.8 | 28.7 | 49.7 | 59.2 |
| CMT-S (Guo et al., 2022a) | Trans | 44.3 | 65.5 | 47.5 | 27.1 | 48.3 | 59.1 |
| Pola-PVT-S (Meng et al., 2025) | Linear | 43.2 | 64.1 | 46.4 | 28.0 | 46.4 | 57.9 |
| NaLaFormer-S | Linear | **47.2** | **68.0** | **50.7** | **29.0** | **51.3** | **63.3** |

*Table 18.* Full table of LRA: Throughput and Peak Memory of various models. A denotes the accuracy, T denotes the throughput of each model and M denotes the peak memory cost.

| | | Softmax | Kernelized | Nystrom | Linformer | Informer | Skyformer | PoLaFormer | NaLaFormer (ours) |
|---|---|---|---|---|---|---|---|---|---|
| Img (1k) | A | 39.14 | 32.63 | 38.94 | 38.43 | 37.86 | 40.77 | 42.15 | 42.54 |
| | T | 736.36 | 862.32 | 1251.28 | 1613.19 | 85.85 | 923.04 | 1340.89 | 1314.03 |
| | M | 9645 | 13013 | 5941 | 3471 | 5357 | 8091 | 4505 | 4211 |
| Path (1k) | A | 70.39 | 69.86 | 69.34 | 65.39 | 56.44 | 70.73 | 70.53 | 71.31 |
| | T | 691.67 | 811.59 | 1125.08 | 1057.03 | 299.94 | 748.98 | 1065.63 | 1292.93 |
| | M | 4831 | 6515 | 2980 | 1745 | 2687 | 4055 | 2286 | 2107 |
| List (2k) | A | 38.71 | 38.46 | 37.95 | 36.44 | 37.05 | 38.69 | 37.35 | 38.21 |
| | T | 402.06 | 496.48 | 834.85 | 528.52 | 305.53 | 627.14 | 949.80 | 802.51 |
| | M | 4473 | 6084 | 1186 | 881 | 2737 | 1712 | 1151 | 2520 |
| Text (4k) | A | 61.55 | 60.02 | 62.36 | 57.29 | 62.13 | 64.7 | 73.06 | 73.48 |
| | T | 252.06 | 327.27 | 1330.68 | 970.90 | 521.16 | 949.80 | 876.74 | 521.49 |
| | M | 17122 | 11720 | 2043 | 1742 | 5736 | 3082 | 1155 | 2102 |
| Retri (4k) | A | 80.93 | 82.11 | 80.89 | 77.85 | 79.35 | 82.06 | 80.5 | 80.42 |
| | T | 116.30 | 144.83 | 496.48 | 424.18 | 142.94 | 348.60 | 344.93 | 207.29 |
| | M | 8947 | 10699 | 2011 | 1649 | 3399 | 2987 | 1139 | 2079 |
| Avg | A | 58.14 | $56.62_{-1.52}$ | $57.90_{-0.24}$ | $55.08_{-3.06}$ | $54.57_{-3.57}$ | $59.39_{+1.25}$ | $60.72_{+2.58}$ | $61.19_{+3.05}$ |
| | T | 439.69 | $528.50_{\times1.20}$ | $1007.68_{\times2.29}$ | $918.77_{\times2.09}$ | $271.08_{\times0.62}$ | $719.51_{\times1.80}$ | $915.60_{\times2.08}$ | $827.65_{\times1.88}$ |
| | M | 9003.6 | $9606.2_{\times1.07}$ | $2832.2_{\times0.31}$ | $1897.6_{\times0.21}$ | $3983.2_{\times0.44}$ | $3985.4_{\times0.44}$ | $2047.2_{\times0.22}$ | $2603.8_{\times0.29}$ |

*Table 19.* Comparison between our method and other SR Models on lightweight image super-resolution. The "LAT" denotes the inference latency and "MEM" represents peak memory usage.

| MODEL | SCALE | SET5 PSNR | SET5 SSIM | SET14 PSNR | SET14 SSIM | BSD100 PSNR | BSD100 SSIM | URBAN100 PSNR | URBAN100 SSIM |
|---|---|---|---|---|---|---|---|---|---|
| Bicubic | ×4 | 28.42 | 0.81 | 26.00 | 0.70 | 25.96 | 0.67 | 23.14 | 0.66 |
| VDSR (Kim et al., 2016) | ×4 | 31.35 | 0.88 | 28.01 | 0.77 | 27.29 | 0.73 | 25.18 | 0.75 |
| MemNet (Tai et al., 2017) | ×4 | 31.74 | 0.89 | 28.26 | 0.77 | 27.40 | 0.73 | 25.50 | 0.76 |
| SRMDNF (Zhang et al., 2018) | ×4 | 31.96 | 0.89 | 28.35 | 0.78 | 27.49 | 0.73 | 25.68 | 0.77 |
| SRFBN-S (Li et al., 2019) | ×4 | 31.98 | 0.89 | 28.45 | 0.78 | 27.44 | 0.73 | 25.71 | 0.77 |
| LAPAR-B (Li et al., 2020) | ×4 | 31.94 | 0.89 | 28.46 | 0.78 | 27.52 | 0.73 | 25.85 | 0.79 |
| ESRN-V (Song et al., 2020) | ×4 | 31.99 | 0.89 | 28.49 | 0.78 | 27.50 | 0.73 | 25.87 | 0.78 |
| ECBSR (Zhang et al., 2021) | ×4 | 31.92 | 0.89 | 28.34 | 0.78 | 27.48 | 0.74 | 25.81 | 0.78 |
| ESRT (Lu et al., 2022) | ×4 | **32.01** | 0.89 | 28.44 | 0.77 | 27.48 | 0.73 | **25.85** | 0.78 |
| NaLaSR | ×4 | 32.00 | **0.89** | **28.50** | **0.78** | **27.49** | **0.73** | 25.83 | **0.78** |

| **Efficiency** | SCALE | LAT | MEM | LAT | MEM | LAT | MEM | LAT | MEM |
|---|---|---|---|---|---|---|---|---|---|
| ESRT (Lu et al., 2022) | ×4 | 195ms | 3.0G | 188ms | 7.0G | 79ms | 2.2G | 195ms | 69G |
| NaLaSR | ×4 | 159ms | 2.3G | 147ms | 2.9G | 72ms | 2.1G | 124ms | 5.3G |
| - SAVE | ×4 | 18.5% | 23.3% | 21.8% | 58.6% | 8.9% | 4.5% | 36.4% | 92.3% |

*Table 20.* Quantitative comparison and efficiency analysis on benchmark datasets for image super-resolution. The best results are highlighted in **bold**. The "LAT" denotes the inference latency and "MEM" represents peak memory usage.

| MODEL | SCALE | SET5 PSNR | SET5 SSIM | SET14 PSNR | SET14 SSIM | BSD100 PSNR | BSD100 SSIM | URBAN100 PSNR | URBAN100 SSIM | MANGA109 PSNR | MANGA109 SSIM |
|---|---|---|---|---|---|---|---|---|---|---|---|
| Bicubic | ×4 | 28.42 | 0.81 | 26.00 | 0.70 | 25.96 | 0.67 | 23.14 | 0.66 | – | – |
| LAPAR-B (Li et al., 2020) | ×4 | 31.94 | 0.89 | 28.46 | 0.78 | 27.52 | 0.73 | 25.85 | 0.79 | – | – |
| ECBSR (Zhang et al., 2021) | ×4 | 31.92 | 0.89 | 28.34 | 0.78 | 27.48 | 0.74 | 25.81 | 0.78 | – | – |
| ESRT (Lu et al., 2022) | ×4 | 32.01 | 0.89 | 28.44 | 0.77 | 27.48 | 0.73 | 25.85 | 0.78 | – | – |
| NaLa-ESRT | ×4 | 32.00 | 0.89 | 28.50 | 0.78 | 27.49 | 0.73 | 25.83 | 0.78 | – | – |
| SwinIR (Liang et al., 2021) | ×4 | 32.44 | 0.8976 | 28.77 | 0.7858 | 27.69 | 0.7406 | 26.47 | 0.7980 | 30.92 | 0.9151 |
| DCTLSA (Zeng et al., 2023) | ×4 | 32.44 | 0.8973 | 28.77 | 0.7846 | 27.67 | 0.7386 | 26.41 | 0.7944 | 31.04 | 0.9138 |
| Nala-DCTLA | ×4 | **32.61** | **0.8992** | **28.91** | **0.7878** | **27.75** | **0.7414** | **26.74** | **0.8037** | **31.43** | **0.9181** |
| Bicubic | ×3 | 30.39 | 0.87 | 27.55 | 0.77 | 27.21 | 0.74 | 24.46 | 0.73 | – | – |
| SRFBN-S (Li et al., 2019) | ×3 | 34.20 | 0.93 | 30.10 | 0.84 | 28.96 | 0.80 | 27.66 | 0.84 | – | – |
| LAPAR-B (Li et al., 2020) | ×3 | 34.20 | 0.93 | 30.17 | 0.84 | 29.03 | 0.80 | 27.85 | 0.85 | – | – |
| ESRN-V (Song et al., 2020) | ×3 | 34.23 | 0.93 | 30.27 | 0.84 | 29.03 | 0.80 | 27.95 | 0.85 | – | – |
| ESRT (Lu et al., 2022) | ×3 | 34.13 | 0.92 | 30.24 | 0.84 | 28.99 | 0.80 | **27.88** | 0.85 | – | – |
| NaLaSR | ×3 | **34.21** | **0.93** | **30.24** | **0.84** | **29.00** | **0.80** | 27.87 | **0.85** | – | – |
| SwinIR (Liang et al., 2021) | ×3 | 34.62 | 0.9289 | 30.54 | 0.8463 | 29.20 | 0.8082 | 28.66 | 0.8624 | 33.98 | 0.9478 |
| DCTLSA (Zeng et al., 2023) | ×3 | 34.76 | 0.9296 | 30.64 | 0.8474 | 29.27 | 0.8093 | 28.81 | 0.8674 | 34.42 | 0.9492 |
| Nala-DCTLA | ×3 | **34.81** | **0.9299** | **30.71** | **0.8486** | **29.32** | **0.8104** | **29.06** | **0.8693** | **34.61** | **0.9504** |

| **Efficiency@RTX3090** | SCALE | LAT | MEM | LAT | MEM | LAT | MEM | LAT | MEM | LAT | MEM |
|---|---|---|---|---|---|---|---|---|---|---|---|
| Nala-DCTLA | ×4 | **32.07ms** | **0.33GB** | **49.30ms** | **0.51GB** | **31.85ms** | **0.28GB** | **155.12ms** | **1.51GB** | **195.34ms** | **1.19GB** |
| SwinIR | ×4 | 594.16ms | 2.67GB | 1297.00ms | 4.20GB | 884.83ms | 1.65GB | 4510.52ms | 12.44GB | 5664.95ms | 9.83GB |
| - SAVE | ×4 | 94.60% | 87.64% | 96.20% | 87.86% | 96.40% | 83.03% | 96.56% | 87.86% | 96.55% | 87.89% |
| DCTLSA | ×4 | 57.70ms | 1.05GB | 109.04ms | 1.47GB | 72.06ms | 0.59GB | 349.86ms | 4.28GB | 456.71ms | 3.44GB |
| - SAVE | ×4 | 44.42% | 68.57% | 54.79% | 65.31% | 55.80% | 52.54% | 55.67% | 64.72% | 57.23% | 65.41% |
| Nala-DCTLA | ×3 | **32.56ms** | **0.43GB** | **52.03ms** | **0.65GB** | **35.65ms** | **0.31GB** | **172.56ms** | **1.96GB** | **218.68ms** | **1.55GB** |
| SwinIR | ×3 | 625.78ms | 2.65GB | 1330.60ms | 4.16GB | 869.78ms | 1.64GB | 4511.03ms | 12.34GB | 5662.68ms | 9.75GB |
| - SAVE | ×3 | 94.80% | 83.77% | 96.09% | 84.38% | 95.90% | 81.10% | 96.17% | 84.12% | 96.14% | 84.10% |
| DCTLSA | ×3 | 57.90ms | 1.66GB | 113.04ms | 2.27GB | 76.74ms | 0.93GB | 355.81ms | 7.19GB | 571.86ms | 5.49GB |
| - SAVE | ×3 | 43.77% | 74.10% | 53.97% | 71.37% | 53.54% | 66.67% | 51.50% | 72.74% | 61.76% | 71.77% |

*Table 21.* Full table of the comparison of the ImageNet-1K classification with the SOTA efficient vision models. The PARA" column denotes the number of model parameters, the FLOPs" column represents the computational amount, and the "ACC" (%) column indicates the top-1 accuracy.

| MODEL | PARA | FLOPs | ACC |
|---|---|---|---|
| LocalVim-T (Huang et al., 2024) | 8M | 1.5G | 76.2 |
| MetaLA (Chou et al., 2024) | 6M | - | 75.3 |
| Mambaout-F (Yu & Wang, 2025) | 7M | 1.2G | 78.9 |
| EfficientVMamba-S (Pei et al., 2025) | 11M | 1.3G | 78.7 |
| NaLaFormer-XT | 8M | 1.0G | **79.1** |
| VAN-b1 (Guo et al., 2022c) | 14M | 2.5G | 81.1 |
| Conv2Former-N (Hou et al., 2024) | 15M | 2.2G | 81.5 |
| SBCFormer-L (Lu et al., 2024b) | 19M | 2.7G | 81.1 |
| RMT-T (Fan et al., 2024) | 14M | 2.5G | 82.4 |
| Agent-PVT-T (Han et al., 2024c) | 12M | 2.0G | 78.4 |
| NaLaFormer-T | 15M | 2.7G | **82.6** |
| Conv2Former-T (Hou et al., 2024) | 27M | 4.4G | 83.2 |
| MambaOut-T (Yu & Wang, 2025) | 27M | 4.5G | 82.7 |
| MogaNet-S (Li et al., 2024) | 25M | 5.0G | 83.4 |
| InternImage-T (Wang et al., 2023) | 30M | 5.0G | 83.5 |
| VMamba-T (Liu et al., 2024) | 30M | 4.9G | 82.6 |
| LocalVMamba-T (Huang et al., 2024) | 26M | 5.7G | 82.7 |
| SG-Former-S (Ren et al., 2023) | 23M | 4.8G | 83.2 |
| MOAT-0 (Yang et al., 2023) | 28M | 5.7G | 83.3 |
| Pola-Swin-T (Meng et al., 2025) | 29M | 4.5G | 82.6 |
| ViG-H-T (Liao et al., 2025) | 29M | 4.5G | 82.8 |
| MILA-T (Han et al., 2024b) | 25M | 4.2G | 83.5 |
| RAVLT-S (Fan et al., 2025b) | 26M | 4.6G | 84.2 |
| NaLaFormer-S | 26M | 5.1G | **84.3** |
| MambaOut-S (Yu & Wang, 2025) | 49M | 9.0G | 84.1 |
| VMamba-S (Liu et al., 2024) | 50M | 8.7G | 83.6 |
| StructViT-B-8-1 (Kim et al., 2024) | 52M | 12G | 84.3 |
| SOFT-L (Lu et al., 2024a) | 64M | 11G | 83.1 |
| FLattn-Swin-S (Han et al., 2023) | 51M | 8.7G | 83.5 |
| Agent-Swin-S (Han et al., 2024c) | 50M | 8.7G | 83.7 |
| Pola-Swin-S (Meng et al., 2025) | 50M | 8.7G | 83.6 |
| ViG-H-S (Liao et al., 2025) | 50M | 8.8G | 83.8 |
| NaLaFormer-B | 52M | 12G | **85.2** |
| InterImage-B (Wang et al., 2023) | 97M | 16G | 84.9 |
| MambaOut-S (Yu & Wang, 2025) | 85M | 16G | 84.2 |
| VMamba-B (Liu et al., 2024) | 89M | 15G | 83.9 |
| SG-Former-B (Ren et al., 2023) | 78M | 16G | 84.7 |
| FLatten-Swin-B (Han et al., 2023) | 89M | 15G | 83.8 |
| Agent-Swin-B (Han et al., 2024c) | 88M | 15G | 84.0 |
| Pola-Swin-B (Meng et al., 2025) | 88M | 15G | 83.8 |
| SMT-L (Lin et al., 2023) | 81M | 18G | 84.6 |
| VRWKV-B (Duan et al., 2025) | 94M | 18G | 82.0 |
| RMT-L (Fan et al., 2024) | 95M | 18G | 85.5 |
| InLine-Swin-B (Han et al., 2024a) | 88M | 15G | 82.0 |
| MILA-B (Han et al., 2024b) | 96M | 16G | 85.3 |
| ViG-H-B (Liao et al., 2025) | 89M | 16G | 84.2 |
| RAVLT-L (Fan et al., 2025b) | 95M | 16G | 85.5 |
| NaLaFormer-L | 95M | 18G | **85.7** |

