# OpenReview forum: "Norm$\times$Direction: Restoring the Missing Query Norm in Vision Linear Attention"
_ICML.cc/2026/Conference — ICML 2026 regular_

### Official Review · Reviewer_kMYz · 2026-03-06

**Soundness:** 2
**Presentation:** 3
**Significance:** 2
**Originality:** 3
**Overall Recommendation:** 4
**Confidence:** 3

**Summary:**

The paper identifies two major limitations of linear attention: 1) query-norm unawareness; 2) neglecting negative values.
To mitigate them, the paper according proposes a two-fold methods: 1) integrating the query-norm into the computation of linear attention, thereby restoring the impact of a query's norm; 2) keeping non-negativity with cosine direction, which makes the negative values also influence the direction. Then, the paper conduct extensive experiments to verify the effectiveness and efficiency of their combined method, NaLaFormer.

**Compliance With Llm Reviewing Policy:**

Affirmed.

**Final Justification:**

I believe the field of visual linear attention is heading down a problematic path, where much of the effort is spent on incorporating new tricks to outperform previous methods, while masking the true contribution of the core method itself. Nonetheless, after re-evaluation, I acknowledge that this paper does offer a novel perspective, and it would be unfair to project my broader criticism of the field onto this work.

**Key Questions For Authors:**

1. Ablation study is given in Table 9, and comparison under the same setting is given in Table 10. My question is why the authors did not ablate the impact the DWC module and the gating mechanism? According to my personal experiment under the DeiT setting, removing them makes NaLaFormer performs worse than a bunch of efficient attention methods, while struggling to show clear improvement upon normal linear attention.

2. Considering that QK-Norm has been widely adopted, is restoring the missing query norm in linear attention still a promising direction for improving it?

3. Regarding Figure 3, does it showing that as the query's norm increases, it is more likely to pay attention to a smaller number of tokens (i.e., being more focused, thereby reducing the entropy/uncertainty)? I'm still not sure about whether this phenomenon in standard attention is a desired one. For example, the paper 'ViT Needs Registers' shows that the high-norm outliers abrupt the attention map and lead to inferior performance.

**Limitations:**

See Weaknesses.
I acknowledge that uncovering the query-norm unawareness of linear attention is an interesting and meaningful discovery.
But, I sincerely encourage the author to explore its more faithful impact, instead of competing the accuracy on ImageNet-1K with the help of more additional modules.
It's very hard to give **reject** to this well-written paper with comprehensive experiments, though I find fatal limitations.
Therefore, I give a recommendation of **weak reject** at this stage.

**Strengths And Weaknesses:**

Strengths:
1. The paper is well-written, the motivation is very convincing.
2. The paper conducts extensive experiments, showing better performance over many baseline methods.
3. Figure 3 is extremely enlightening.

Weaknesses:
1. (**Major Weaknesses**) The paper claims that the expressiveness loss of linear attention arise from two primary causes: 1) the unawareness of a query's norm; 2) discarding the negative values. This implies that, given the two issues mitigated, the performance of linear attention can match that of standard attention. However, the reviewer notice that: 1) the performance gain of NaLaFormer mainly comes from two additional modules, the depth-wise convolution and the data-dependent gating matrix. With them removed, NaLaFormer performs not very well (I actually conduct an experiment to find out this; I truly appreciate the authors' integrity for submitting their code).

2. (**Major Weaknesses**) Uncovering linear attention's unawareness of a query's norm is the central contribution and motivation of the paper. However, in many LLMs, QK-Norm is applied, which wipes out the impact of norm. Moreover, a recent advance of ViT, ViT-5, has validated that using QK-Norm for standard attention is beneficial. This suggests that norm awareness is not a very crucial component of the high expressiveness of standard attention.

---

> ### Author Rebuttal · Authors · 2026-03-31
>
> Appendix: https://shorturl.at/K31Is
>
> >W1. Q1. Impact of DWC and gating mechanism.
>
> We sincerely thank the reviewer for this important concern, and we especially appreciate the reviewer’s effort in checking our released code. We agree that this point should be isolated more carefully.
> Our intention is not to attribute the gain to DWC or gating. These modules are common default components in recent vision linear-attention backbones:
> since FLatten Transformer[1], DWC has been widely adopted in efficient vision attention models, considered as a residual branch or a LEPE-like local enhancement module to inject locality bias. Similar designs appear in a broad range of recent sub-quadratic vision models, including AgentAttn[2], InLine[3], Circulant Attention[4], MLLA[5], LinearDiff[6], RMT[7], RALA[8], and MALA[9]. As for the gating mechanism, its role and cost have also been discussed in prior work. PolaFormer[10] shows that gating has negligible impact on the overall complexity and computation, while providing clear benefits in improving the effective rank and expressiveness of linear attention. In addition, recent DeltaNet-based models such as Qwen3.5 and Kimi Linear[11] also adopt gating, and subsequent works such as Gated Attention[12] further explore this design.wo me Later methods including Circulant Attention[4], MLLA[5], RALA[8], MALA[9], and SAGA[13] also include gating as a default component (the reference table can be found in Appendix).
>
> Therefore, from this perspective, DWC and gating should be understood as common and fair default configurations in this line of research, rather than the main source of the performance gain. But to directly address the reviewer’s concern, we additionally performed a controlled ablation by **removing both DWC and gating**. Under this simplified setting, NaLaFormer still performs best:
>
>
> |Method|Swin|PVT|
> |---|---|---|
> |QKNorm|80.58|75.06|
> |FLatten|80.62|74.91|
> |Ours|80.83|75.41|
>
>
>
>
> >W2. Q2. Norm-aware v.s. QK-norm
>
>
> Thank you for this important comment! We agree that results such as QK-Norm and ViT-5 suggest that raw query/key norms should not dominate attention computation in an *uncontrolled* manner. However, this does not contradict our motivation. Our point is not that larger norms are always better, but that **query norm contains useful information related to attention concentration, and therefore should not be completely removed, but rather used in a controlled way**.
>
> To directly address this concern, we additionally compare NaLaFormer with QK-Norm-based variants. On both **small-scale models** such as **Swin** and **PVT**, as well as the **ViT-Large** setting where QK-Norm is more commonly adopted, NaLaFormer consistently outperforms the corresponding QK-Norm baselines. This suggests that the effectiveness of QK-Norm does not imply that query norm is unimportant; rather, it supports a more nuanced conclusion that **norm should be controlled, rather than completely removed**. We will include these additional results and clarify this point in the revised version.
>
>
> >Q3. Interpretation of Figure 3 and relation to high-norm outliers.
>
> Thank you for this insightful question! We would like to clarify the interpretation of Fig. 3 more precisely. The figure **does not show** that the model attends to only a few tokens, nor that high-q-norm queries are rare outliers. In fact, the high-q-norm region is populated by **many tokens**.
>
> What Fig. 3 shows is that the **correlation between query norm and attention entropy is better aligned**. This is a query-side correlation analysis, rather than evidence that the model globally selects only a few tokens as attention targets.
>
> We agree with the reviewer that uncontrolled high-norm outliers can be harmful, which is also consistent with ViT Needs Registers. This is exactly why NaLaFormer does *not* feed raw query norm back into attention directly. Instead, it restores query-side norm information only through a **bounded smooth mapping**, so that the useful modulation of attention concentration is preserved **without allowing extreme values to dominate**. We will clarify this interpretation more explicitly in the revision.
>
> We hope the above clarifications address your concerns. Please feel free to let us know if you have any further questions or suggestions!

---

> > ### Author Rebuttal · Reviewer_kMYz · 2026-04-02
> >
> > Since the authors provide only limited experiments and comparisons, it is still difficult for me to assess the impact of restoring query normalization versus not, as well as the impact of gating. Depth-wise convolution (DWC) has been adopted in many prior methods, but among the compared baselines, I only found this modification in PoLaFormer. Moreover, according to the ablation study in PoLaFormer, gating appears to play a very important role. Therefore, I still suspect that gating accounts for most of the empirical gains, rather than the core query-norm-aware design itself. For this reason, I maintain my original rating.

---

> > > ### Author Response · Authors · 2026-04-03
> > >
> > > Thank you for your kind reply and for the valuable time and effort you devoted to carefully reviewing our paper. For the points that may still seem insufficiently addressed, we would like to offer the following additional clarification.
> > > > Impact of DWC, Gating, and query-norm restoration of NalaFormer.
> > >
> > > We would like to kindly clarify that there may be a misunderstanding here due to the limited space in the main paper. Table 1 only presents a compact subset of the ImageNet-1K comparisons, while **the full comparison is provided in Table 17**. In the full table, many recent vision linear attention baselines already adopt DWC and/or gating modules as part of their standard design. Specifically, beyond PolaFormer, methods such as *FLatten Transformer[[Link](https://shorturl.at/X0OgP)], RMT[[Link](https://shorturl.at/eqfBh)], AgentAttn[[Link](https://shorturl.at/HgaPo)], RAVLT[[Link](https://shorturl.at/c7zfv)], MILA[[Link](https://shorturl.at/7tASO)], and InLine [[Link](https://shorturl.at/DxDyY)]* all employ DWC modules, while *ViG[[Link](https://shorturl.at/NJjWW)] and RAVLT[[Link](https://shorturl.at/B6DPX)]* also adopt gating modules in their implementations. Moreover, several more recent works, including *Circulant Attention[[Link](https://shorturl.at/CjxOx)], MALA[[Link](https://shorturl.at/FYZDr)], and Linear Diff[[Link](https://shorturl.at/bDpor)],* also follow this design choice by incorporating DWC and gating. Therefore, these components are not special additions unique to our method, but are already common practices in recent vision linear attention design. Our implementation follows this established practice and we will **explicitly emphasize** that our model adopts both modules in our revised version.
> > >
> > > We would also like to note that not all of our experiments rely on DWC. Beyond the classification results, neither the original ESRT-based SR experiments nor our additional DCTLA-based results use DWC. In diffusion tasks, we do not use DWC for DiT, while for SiT we retain DWC only because it is already part of the original baseline architecture. In the LLM experiments, we do not use DWC either, and instead directly replace the kernel function. These results suggest that the effectiveness of our method is not tied to this module.
> > >
> > > Overall, we hope that the above clarifications help more clearly disentangle the effect of the proposed query-norm-aware design from auxiliary modules. ***We would sincerely appreciate your thoughtful consideration of this additional evidence in your final assessment.*** Thanks again for your time and attention.

---

### Official Review · Reviewer_7cUV · 2026-03-08

**Soundness:** 3
**Presentation:** 3
**Significance:** 3
**Originality:** 3
**Overall Recommendation:** 4
**Confidence:** 4

**Summary:**

This paper identifies two key limitations of existing linear attention mechanisms. The first is that the L1 normalization inherent to Linear Attention annihilates the query norm, breaking the query-norm–entropy correlation that Softmax Attention naturally preserves. The second is that standard non-negativity enforcement via ReLU or power functions eliminates valid inner-product interactions between query-key pairs with opposing directions. This paper proposes NaLaFormer, a Norm-Aware Linear Attention for Transformer models, which built upon a norm × direction(ND) decomposition of the query and key vectors to solve the two limitations. The query norm is re-injected into a kernel power function to modulate attention spikiness; the direction vectors are processed via a cosine-based similarity metric that enforces non-negativity while retaining the sign-sensitive geometry of the original inner product. This paper conducts extensive theoretical analysis on the role of query specification in attention entropy, and empirical evaluation shows that it improves performance in visual and language model benchmarks.

**Compliance With Llm Reviewing Policy:**

Affirmed.

**Final Justification:**

The rebuttal addressed most of my concerns. The hyperparameter issue remains, but in the big picture it is a minor concern. I'd like to maintain my original positive rating.

**Key Questions For Authors:**

1.	Could you discuss if other recent Linear Attention methods inject query norm explicitly or implicitly?

2.	Could you provide more implementation details for the diffusion experiments, such as the number of training steps, denoising steps, and other training settings used for the DiT/SiT S/2 models?

**Limitations:**

Yes

**Strengths And Weaknesses:**

## Strengths

1. The paper provides a clear theoretical argument and empirical experiment highlight the importance of the q-norm.

2. The design of ND decomposition is concise, clear, and easy to implement.

3. The evaluation covers a wide range of tasks and modalities, including classification, dense prediction, image generation, super-resolution, language modelling and long-range sequence tasks. This breadth of evaluation is commendable.

## Weaknesses

**Limited comparison in the analysis of Figure 3.** The paper analyzes the relationship between query norm and attention entropy by comparing the proposed method with ELU+1 Linear Attention and FLatten. However, these baselines mainly represent early Linear Attention designs. Recent methods like inLine and MetaLA are not included in the analysis.

**Limited justification for query-norm-aware feature map design.** In Section 3.1, the proposed query-norm-aware feature map introduces a scaling function f(∥q∥) applied to the query direction, where the paper chooses f(x)=λ(τ+tanh⁡(x)). However, the justification for this specific functional form appears limited. The paper mainly motivates this design from an engineering perspective (e.g., avoiding numerical overflow and ensuring the exponent remains larger than 1), but does not provide theoretical analysis explaining why this formulation is preferable to other monotonic functions of the query norm.

Furthermore, the ablation results suggest that the model performance can be sensitive to the hyperparameters λ and τ, particularly in the image experiments. This raises the question of whether the proposed formulation is inherently robust, or whether the performance gains depend on careful hyperparameter tuning. Additional analysis or broader ablations over alternative functional forms could help clarify the necessity of this design choice.

**Baselines for the memory-efficiency claim.** The 92.3% memory reduction claim compares NaLaSR against ESRT, a softmax-based transformer. Ideally, the comparison should be against other local-attention SISR models (e.g., SwinIR, HAT).

**Minor issues.**
- The meanings of the columns of Fig. 1 are not explained.

---

> ### Author Rebuttal · Authors · 2026-03-31
>
> Appendix: https://shorturl.at/tP1Bv
>
> >W1. Limited comparison in Figure 3.
>
> Thanks! We agree that comparing only earlier linear-attention variants is not sufficient. To address this, we added InLine to the query-norm–entropy analysis (**see attachement in the link**) in the same spirit as Fig. 3. The same overall trend remains: preserving query-scale information helps recover the attention-concentration behavior that standard linear attention loses. We were unable to include MetaLA because its vision checkpoint/training code is not publicly available, and reproducing the vision model from scratch is beyond the rebuttal budget. We will clarify this limitation and include the added InLine result in the revision.
>
>
> >W2. Justification for query-norm-aware feature map
>
> Thank you for your thoughtful comment and agree that the functional form deserves further clarification. From both theoretical motivations (Theorem A.2,3) and practical considerations, feature map design requires two properties:
> 1. **Monotonicity**, so the larger query norms lead to sharper attention distributions.
> 2. **Boundedness**, so the exponent remains stable.
>
> The tanh function naturally **satisfies both**: it is monotonic, bounded, and smooth. Its asymptotic behavior (linear near zero, saturating to a constant) provides a controlled way to scale the exponent without causing extreme values, which is essential for stable training.
>
> Other monotonic functions (e.g., sigmoid, clipped linear functions) could also be viable. The tanh formulation was chosen for its simplicity. We do not claim it is uniquely optimal; rather, we emphasize that the key contribution is the principle of making the feature map query-norm-aware, and the function serves as a concrete instantiation of this principle.
>
>
>
> >W3. Sensitivity of λ and τ
>
> We agree that Table 12 shows some sensitivity to hyperparams in vision tasks. In vision tasks, query and key vectors often exhibit larger norm variation across image patches, so $\lambda$, which modulates attention entropy, more directly affects the modeling of spatial patterns. Meanwhile, $\tau$ controls the preservation of directional information and numerical stability. We would like to emphasize that $\lambda$ and $\tau$ are designed to restore the sharpness modulation effect of query norm on attention entropy in softmax attention. Therefore, their roles are well motivated by the formulation itself, rather than being introduced solely for careful tuning.
>
> In addition, in Table 12, the NLP Retrieval task uses a sequence length of 4K, whereas the Image task uses only 1K. In linear attention, shorter sequences make the contribution of each individual token to the shape of the attention distribution more direct, and the model therefore tends to be more sensitive to $\lambda$ and $\tau$.
>
>
> >W4. The comparison should be against other local-attention SISR models
>
> Thank you for the suggestion. For a fair comparison, we choose SwinIR and a SwinIR-based linear model DCTLA [1] as the baseline, and report not only its performance but also its efficiency on an RTX 3090. Please refers to **full table presented in the links**.
>
>
>
> [1] "Densely connected transformer with linear self-attention for lightweight image super-resolution." IEEE Trans. Instrumentation and Measurement (2023).
>
>
> >W5. Meanings of the columns of Fig. 1
>
> Thank you for pointing this out. **Each column corresponds to the attention statistics from the same image**, and is used to illustrate the relationship between **attention entropy and the query/key norm** for that image. Specifically, the top row shows the relationship between **entropy and q-norm** for that image; the bottom row shows the relationship between **entropy and k-norm** for that image. Therefore, the columns simply correspond to visualizations from different image samples. We will clarify this in the revised version to avoid ambiguity.
>
> >Q1. Could you discuss if other recent Linear Attention methods inject query norm explicitly or implicitly?
>
> Thank you for the question. We believe several recent methods already preserve query-norm information, either explicitly or implicitly. For example, Linear log-normal attention does so more explicitly, since its exponential feature map preserves magnitude information and affects concentration behavior. InLine does so more implicitly, by redesigning normalization to avoid losing scale-related information. Our work differs in that we directly connect query norm to **attention-entropy modulation**, and use this as the motivation for feature-map design. This is the same core content as your draft, but in a much cleaner taxonomy form.
>
>
>
> >Q2. Implementation details for the diffusion
>
> Thank you for your question. For the diffusion experiments, we followed the official DiT and SiT and replaced only the original attention module with NaLaFormer. All training hyperparameters were kept unchanged. All other components remain identical to the original implementations.

---

> > ### Author Rebuttal · Reviewer_7cUV · 2026-04-03
> >
> > I thank the authors for the response and they answered most of my questions. It seems to me the hyperparameter sensitivity remains an issue, albeit not a major one. I would like to maintain my current score, which is positive.

---

> > > ### Author Response · Authors · 2026-04-07
> > >
> > > Dear Reviewer 7cUV,
> > >
> > > Thank you very much for your kind review and helpful suggestions. We are glad that our rebuttal has satisfactorily addressed your concerns. We will incorporate the additional results and corresponding discussion into the revised version. Thank you again for your time and careful consideration.
> > >
> > > Best,
> > >
> > > Authors of Submission 953

---

### Official Review · Reviewer_HmcA · 2026-03-10

**Soundness:** 3
**Presentation:** 3
**Significance:** 3
**Originality:** 3
**Overall Recommendation:** 5
**Confidence:** 3

**Summary:**

This work addresses the limited expressiveness of linear attention and introduces an analysis perspective based on “Norm×Direction” (ND) decomposition, clearly identifying that performance degradation mainly arises from the suppression of query norms during normalization and the loss of directional information due to non-negativity constraints. To address these issues, NaLaFormer is proposed, which combines a query-norm-aware feature mapping with a trigonometry-based directional similarity method, effectively enhancing the expressive power of linear attention. Experiments across multiple vision and language tasks demonstrate strong generalization and significant efficiency improvements.

**Compliance With Llm Reviewing Policy:**

Affirmed.

**Final Justification:**

The authors’ response has addressed my concerns, and I will maintain my score.

**Key Questions For Authors:**

Please refer to the Weaknesses section for detailed questions.

**Limitations:**

yes

**Strengths And Weaknesses:**

Strengths
1.Clear problem diagnosis: This paper leverages ND decomposition to reveal that linear attention’s lack of query-norm awareness leads to uncontrollable attention entropy, and theoretically and visually validates the negative correlation between query norm and entropy.
2.Technical innovation: A query-norm-aware sharpening function is proposed to restore control over attention entropy, while a directional mapping based on Ptolemy’s theorem and trigonometric functions preserves inner-product directional information under non-negativity.
3.Comprehensive experimental: The method is validated across vision, generative, language, and long-sequence tasks, showing consistent performance gains.
4.Efficiency gains: Peak memory is reduced by up to 92.3% in high-resolution tasks, demonstrating applicability to large-scale and resource-constrained scenarios.

Weaknesses
1.Limited theoretical discussion: Equations (8)-(10) show NaLaFormer’s dimension-wise cosine mapping differs from Cosformer/RoPE’s positional mechanism, but the paper only discusses this superficially.
2.Computational cost unclear: The dimension-wise cosine computation in Equation (10) may increase overhead; a discussion of its complexity and impact on efficiency is recommended.
3.Missing Softmax comparison: Table 6 shows NaLaFormer outperforms Softmax on irrelevant-signal suppression, but equivalence with Softmax’s exponential mechanism is unproven. A controlled comparison is recommended.
4.Hyperparameter conflict (Table 12): visual tasks are sensitive to the norm-sharpening parameter, unlike language tasks, but this is not analyzed.
5.Power function necessity: The query-norm-aware module uses f(x) =λ*(τ+ tanh(x)), but it is unclear if simple linear scaling (f(x)=λx) would yield similar results. Ablation experiments comparing different norm-injection forms are recommended.

---

> ### Author Rebuttal · Authors · 2026-03-31
>
> >W1.Limited theoretical discussion
>
> Thanks for the insightful question. As noted in line 263 left, the cosine terms in Cosformer and RoPE are primarily introduced for **positional modeling**, and their effects depend on the relative or absolute positions of tokens (**fixed**). In contrast, the cosine mechanism in NaLaFormer is designed to characterize the **directional similarity** between queries and keys (**learned**), where the input is derived from the feature vectors themselves rather than positional indices. Building on Ptolemy’s theorem, our method decomposes the inner product into norm and directional components, where the cosine term serves to preserve the relations between unit vectors while maintaining non-negativity. This helps alleviate the overly aggressive suppression of negatively correlated information that often occurs in conventional ReLU-based linear attention.
>
> >W2.Computational cost unclear
>
> We agree that the practical overhead should be quantified explicitly. We therefore measured training cost on the LRA text task for feature dimensions d and 2d, with all other settings fixed (2000 steps, batch size 32, sequence length 4096). Going from d to 2d increases memory from 2000M to 2193M and wall-clock time from 77s to 84s, corresponding to **only +9.7% memory** and **+9.1% time**. We will include this analysis in the revised paper.
>
> >W3. Missing Softmax comparison
>
> We thank the reviewer for this question. We would like to clarify that NaLaFormer is not mathematically equivalent to Softmax attention. Our core claim is instead that NaLaFormer functionally restores the negative correlation between query norm and attention sharpness (or entropy), which is one of the key properties of Softmax attention. In conventional ReLU-based linear attention, this connection is often broken, so the query norm can no longer effectively control the concentration of the attention distribution. By contrast, NaLaFormer re-establishes this correlation through our norm-aware design. Therefore, what we emphasize is a preservation at the functional and behavioral level, rather than a strict equivalence at the operator level.
>
> >W4.Hyperparameter conflict (Table 12)
>
> Thanks! Visual tasks, such as ImageNet classification, rely heavily on local and spatially structured features, where query and key vectors often exhibit substantial norm variation across different image patches. In this setting, the sharpening parameter $\lambda$ directly affects the entropy of the attention distribution, as we theoretically show in Appendix A.1. A larger $\lambda$ amplifies the weights of highly similar pairs, which is particularly important for capturing fine-grained spatial patterns. In contrast, language tasks involve longer sequences and more uniformly distributed token embeddings, making norm modulation relatively less critical. This observation is also consistent with prior linear-attention literature [1,2], which similarly reports that vision models are generally more sensitive to sharpening parameters and local information.
>
> In addition, in Table 12, the NLP Retrieval task uses a sequence length of 4K, whereas the Image task uses only 1K. In linear attention, shorter sequences make the contribution of each individual token to the shape of the attention distribution more direct, and the model therefore tends to be more sensitive to $\lambda$ and $\tau$.
>
> [1] Han, Dongchen, et al. "Bridging the divide: Reconsidering softmax and linear attention." NIPS 2024.
> [2] Meng, Weikang, et al. "PolaFormer: Polarity-aware Linear Attention for Vision Transformers." ICLR 2025.
>
>
> >W5.Power function necessity
>
> Regarding the necessity of the power function design in the query-norm-aware module, we agree that the existing textual explanation alone is insufficient to fully justify why a form such as $f(x)=\lambda(\tau + \tanh(x))$ is needed, rather than a simpler linear scaling $f(x)=\lambda x$. Our design motivation is that the introduction of tanh enables bounded control over the exponent, thereby mitigating the numerical instability or overflow issues that may arise under large norm conditions, while also providing smoother nonlinear modulation. In contrast, although the simple linear function is more direct in form, it tends to cause numerical overflow and instability during training in the large-norm regime due to the power function.
>
> Thank you for these valuable suggestions. We will incorporate the additional analyses into the revised manuscript to better clarify the motivation and design choices. We hope the above clarifications address your concerns. Please feel free to let us know if you have any further questions or suggestions!

---

> > ### Author Rebuttal · Reviewer_HmcA · 2026-04-01
> >
> > The authors’ response has addressed my concerns, and I will maintain my score.

---

> > > ### Author Response · Authors · 2026-04-03
> > >
> > > Dear Reviewer HmcA,
> > >
> > > Thank you very much for your thoughtful feedback and constructive suggestions. We are delighted that our rebuttal has addressed your concerns and we will incorporate the new results and discussion into the revised version. Thanks again for your time and attention.
> > >
> > > Best,
> > >
> > > Authors of Submission 953

---

### Official Review · Reviewer_XLnm · 2026-03-12

**Soundness:** 2
**Presentation:** 3
**Significance:** 3
**Originality:** 3
**Overall Recommendation:** 4
**Confidence:** 5

**Summary:**

The paper addresses "entropy collapse" in linear attention, specifically identifying query norm cancellation as a primary driver of performance degradation. The authors introduce NaLaFormer, which utilizes a Norm×Direction (ND) decomposition. This approach re-injects query magnitude via a power function to restore "spikiness" and employs a trigonometric mapping to ensure non-negativity without information loss. NaLaFormer is validated across an impressive array of tasks—from ImageNet classification to diffusion generation and language modeling—demonstrating consistent gains over linear attention baselines.

**Compliance With Llm Reviewing Policy:**

Affirmed.

**Final Justification:**

My concerns are addressed. Please make sure adding the experiment results shown in the rebuttal into your revised version --- the conslusion can be substantially different without these results.

**Key Questions For Authors:**

1. Given that many foundation models use QK-Norm to suppress norm information for stability, how does NaLaFormer behave as depth and width increase?

2. Have you tested this on a 1B+ parameter regime?

3. What is the performance of a baseline where you simply apply RMSNorm to Q (discarding the norm) while keeping your directional component?

4. Is there a heuristic for setting $\lambda$ and $\tau$, or must they be tuned per task/scale?

5. Wall-clock Efficiency: Can you provide a throughput (tokens/sec) comparison that accounts for the $2d$ dimension expansion in the cosine mapping?

**Limitations:**

The paper overlooks the potential for training instability at scale—a known issue with norm-sensitive attention. Additionally, the task-specific tuning required for new hyperparameters $(\lambda, \tau)$ and the increased constant-factor overhead of the feature map are under-discussed, limiting the "plug-and-play" claims for large-scale deployment.

**Strengths And Weaknesses:**

**Strengths**

1. The ND decomposition provides a mathematically elegant diagnosis of why linear attention fails to match softmax. The connection between query norms and Positive Sequence Entropy (Theorems A.2 & A.3) is well-derived and insightful.

2. The proposed feature maps are "drop-in" replacements, requiring minimal architectural changes. The plug-in evaluation on FLatten-Swin-T effectively isolates the benefits of the attention mechanism itself.

3. Good testing results across vision, language, and generative tasks (including a 92% memory reduction in super-resolution) compared to existing models.

**Weaknesses**

1. My major concern is that the central thesis—that amplifying query norms is beneficial—contradicts modern large-scale training stability practices. In LLMs and deep ViTs, query norms often grow pathologically, leading to instability; hence the industry standard of QK-Norm (e.g., RMSNorm) to discard norm information. The paper lacks evidence that its "norm-aware" approach remains beneficial (or even trainable) at scales beyond 100M parameters.

2. All primary vision experiments stay below ~95M parameters, and the 340M language model is trained on a mere 15B tokens. At these "small-to-medium" scales, performance gains are often noisy. The absence of scaling curves or experiments on larger architectures (e.g., 1B+ language or 200M+ vision) leaves the practical relevance of the method unverified.

3. Missing QK-Normalization Baselines: Since the paper focuses on query norm pathology, a crucial baseline is missing: applying RMSNorm/LayerNorm to Q and K before the feature map. Without this, it is unclear if "norm-awareness" is superior to simply "norm-normalization," which is the standard solution to the problems described.

4. The cosine direction mechanism relies on $\lambda$ and $\tau$, which show significant sensitivity in Table 12 (up to 3 points variance). Furthermore, the mapping doubles the feature dimension ($d$ to $2d$), increasing the constant factor of the $O(N \cdot d^2)$ complexity, yet the actual wall-clock training overhead is not quantified.

5. Weak Language Modeling Significance: The reported gains (+0.5 to +0.7 accuracy) are marginal for 340M models. Without perplexity comparisons on standard benchmarks or evaluations against full softmax attention, the claim that NaLaFormer "surpasses Mamba" feels premature.

---

> ### Author Rebuttal · Authors · 2026-03-31
>
> >W1 and Q1. amplifying query norms is beneficial—contradicts modern practices; evidence of "norm-aware" at scales beyond 100M param.
>
> Thank you for the question. We agree with the motivation behind QK-Norm: raw query/key magnitudes should not directly dominate attention, since *unbounded* logits can cause saturation and instability. Our claim is **not** that larger query norms are always beneficial; rather, our point is that **completely canceling query magnitude** removes a useful query-side signal related to attention concentration.
> NaLaFormer does **not** restore raw norm domination. It first computes similarity from the **normalized** directional component of the query, and reintroduces magnitude only through a **bounded smooth function** $f(||q||)=\lambda(\tau+\tanh(||q||))$ . Therefore, it *cannot recreate the unbounded logit amplification* that QK-Norm is designed to prevent. In this sense, NaLaFormer is aligned with the stabilizing philosophy of QK-Norm, while preserving useful query-side calibration information.
>
> To directly address the scaling/stability concern, we additionally ran **ViT5-L (304M)** with the same replacement strategy as in the main paper. The model trains stably without divergence, and the tracked query norms **remain bounded** throughout training (**4.14–4.91** over epochs 50–300, https://shorturl.at/cdEnc), rather than exhibiting pathological growth. Moreover, ViT5-L reaches **82.6** top-1 on ImageNet, outperforming plain ViT-L/16 (79.66, 304.3M params) at a similar model scale. We will include these ViT5-L stability results and query-norm statistics in the revised paper.
>
> >W2. Q2.Evidence beyond 100M / 1B+ language models
>
> We agree that larger-scale evidence is important. At present, our strongest additional validation is on ViT5-L (304M), which is more directly aligned with the main focus of this work, namely **vision linear attention**. We are also extending the study toward 1B-scale language models, but we prefer not to overclaim beyond the evidence currently available during the rebuttal period (still running).
> Accordingly, in the revised paper we will (i) include the larger-scale vision validation now available, and (ii) revise the wording to avoid implying that NaLaFormer has already been fully validated at frontier LLM scale.
>
> >W3. Q3. Missing QK-Normalization Baselines.
>
> We agree that **QK-Norm is an important baseline** for this work. Following the reviewer’s suggestion, we **added a direct comparison** under the same backbone and training setting. On Swin, QKNorm / FLatten / Ours achieve 80.58 / 80.62 / 80.83, respectively; on PVT, they achieve 75.06 / 74.91 / 75.41, respectively. Compared with the FLatten baseline, QKNorm brings **mixed changes** (-0.04 on Swin and +0.15 on PVT), whereas NaLaFormer consistently **improves** over FLatten on both backbones (+0.21 on Swin and +0.50 on PVT). These results show that the gain is not explained by normalization alone, but comes from our proposed bounded restoration of useful query-side norm information. We will include this QKNorm baseline in the revised paper.
>
> >W4.1. Q4.Heuristic for $\lambda$ and $\tau$?
>
> Thank you for the question. Our empirical observation is that $\lambda$ and $\tau$ do **not** require heavy tuning for each task or scale. In practice, a shared default setting works well in most cases, while vision tasks are somewhat more sensitive, especially to $\lambda$, and may benefit from light validation-based adjustment. We further observe that **longer sequences often favor a larger $\lambda$**, since stronger sharpening helps maintain a sufficiently concentrated attention distribution. We will clarify this heuristic in the paper.
>
> >W4.2. Q5 . Wall-clock overhead of doubling the feature dimension
>
> We agree that the practical overhead should be quantified explicitly. We therefore measured training cost on the LRA text task for feature dimensions d and 2d, with all other settings fixed (2000 steps, batch size 32, sequence length 4096). Going from d to 2d increases memory from 2000M to 2193M and wall-clock time from 77s to 84s, corresponding to **only +9.7% memory** and **+9.1% time**. We will include this analysis in the revised paper.
>
> >W5. Language-modeling significance and the “surpasses Mamba” claim
>
> Thank you for this important comment. We agree that “surpasses Mamba” is too broad and will revise it. Table 7 already reports standard LM perplexity, not only downstream accuracy. Compared with DeltaNet, NaLa+DN improves WikiText from **29.08 to 27.82**, LAMBADA from **50.87 to 49.77**, and average downstream accuracy from **43.6 to 44.3**. Compared with Gated DeltaNet, NaLa+GDN improves WikiText from **26.59 to 25.89** and average downstream accuracy from **44.3 to 44.8**. We therefore position the language results more carefully as evidence that NaLaFormer transfers stably to linear-attention language models and consistently improves DeltaNet-family baselines, rather than claiming broad superiority over Mamba.

---

> > ### Author Rebuttal · Reviewer_XLnm · 2026-04-02
> >
> > My concerns are addressed. Please make sure adding the experiment results shown in the rebuttal into your revised version --- the conslusion can be substantially different without these results.
> >
> > I'd ike to raise my score to 4 if the authors can include new results in the paper.

---

> > > ### Author Response · Authors · 2026-04-03
> > >
> > > Dear Reviewer XLnm,
> > >
> > > Thank you very much for your thoughtful feedback and positive response. We are glad that our rebuttal has addressed your questions and concerns, and we sincerely appreciate your raising the score. Your helpful comments and suggestions have strengthened our work. We will make sure to incorporate all the new results into the revised version, together with the necessary analysis and discussion. Thank you again for your valuable feedback.
> > >
> > > Best,
> > >
> > > Authors of Submission 953

---

### Decision · Program_Chairs · 2026-04-30

**Decision:**

Accept (regular)

**Comment:**

Linear attention is an important and popular topic for deep learning computation: the contributed norm x direction decomposition of attention is novel and useful in vision and language experiments. This work identifies two issues, nullification of the query norm and issues constraining to non-negativity, and the proposed NaLaFormer addresses these directly and results in gained accuracy.

All reviewers agree on acceptance after the rebuttal and responses with 2/4 raising their scores from weak reject to weak accept. XLnm is satisfied by the rebuttal clarifications and results on the query norm, scaling, adding baselines for more normalization, and the computation and claims made. HmcA is positive given the rebuttal on computation, theoretical discussion, missing comparison, and hyperparameters. 7cUV is satisfied with the rebuttal w.r.t. justification of normalization, hyperparameters, comparison to more local attention methods, and clarifies the role of the query norm in other recent attention alternatives. kMYz maintains their borderline accept rating after the rebuttal on the ablation and analysis of the method.

Note: reviewer kMYz had criticisms of the broader topic, which they later recognized as an issue for the topic and not for this work in particular, and so they acknowledged the contributions of the submission.